# Comparative genomics reveals a unique nitrogen-carbon balance system in Asteraceae

Fei Shen [1,5], Yajuan Qin [1,5], Rui Wang [2,5], Xin Huang [1], Ying Wang[1,3], Tiangang Gao[4], Junna He [2], Yue Zhou [3], Yuannian Jiao [4], Jianhua Wei[1] ✉, Lei Li [3] ✉ & Xiaozeng Yang [1] ✉

The Asteraceae (daisy family) is one of the largest families of plants. The genetic basis for its high biodiversity and excellent adaptability has not been elucidated. Here, we compare the genomes of 29 terrestrial plant species, including two de novo chromosome-scale genome assemblies for stem lettuce, a member of Asteraceae, and *Scaevola taccada*, a member of Goodeniaceae that is one of the closest outgroups of Asteraceae. We show that Asteraceae originated ~80 million years ago and experienced repeated paleopolyploidization. PII, the universal regulator of nitrogen-carbon (N-C) assimilation present in almost all domains of life, has conspicuously lost across Asteraceae. Meanwhile, Asteraceae has stepwise upgraded the N-C balance system via paleopolyploidization and tandem duplications of key metabolic genes, resulting in enhanced nitrogen uptake and fatty acid biosynthesis. In addition to suggesting a molecular basis for their ecological success, the unique N-C balance system reported for Asteraceae offers a potential crop improvement strategy.

Angiosperms (flowering plants) experienced a rapid terrestrial radiation and diversification of species, eventually becoming ecologically dominant before the end of the Cretaceous period, famously characterized by Charles Darwin as "an abominable mystery"[1]. In particular, the Asteraceae rival the Orchidaceae as the largest family of flowering plants. The Asteraceae comprise more than 1620 genera and 30,000 species and account for approximately 10% of all flowering species (Supplementary Fig. 1 and Supplementary Note 1)[2,3]. The richness of Asteraceae species is much greater than that of related families in the order Asterales, including Calyceraceae (47 spp.), Goodeniaceae (430 spp.), and Menyanthaceae (60 spp.)[3]. As the most ecologically successful family with incredible diversity and excellent adaptability,

its members occur in nearly every type of habitat on earth, including extreme environments, such as deserts and salt flats (Supplementary Fig. 1)[4]. Another example that can illustrate the extraordinary adaptability of the Asteraceae is that the plants in this family rank among the top three on the list of globally invasive species (Supplementary Fig. 1). The ecological success of Asteraceae is considered to be related to its specific morphology and physiology. For example, the characteristic inflorescence (*capitulum*) substantially contributes to ecological radiation by attracting insect pollinators that rely heavily on this family to feed and reproduce[5]. The achene-like fruits (*cypselae*) with pappus of bristles promote dispersal by wind or attach to the fur or plumage of animals[2]. Both of these methods of dispersion result in seeds that

[1]Beijing Key Laboratory of Agricultural Genetic Resources and Biotechnology, Institute of Biotechnology, Beijing Academy of Agriculture and Forestry Sciences, 100097 Beijing, China. [2]Beijing Key Laboratory of Development and Quality Control of Ornamental Crops, College of Horticulture, China Agricultural University, 100193 Beijing, China. [3]State Key Laboratory of Protein and Plant Gene Research, School of Advanced Agricultural Sciences, Peking University, 100871 Beijing, China. [4]State Key Laboratory of Evolutionary and Systematic Botany, Institute of Botany, the Chinese Academy of Sciences, 100093 Beijing, China. [5]These authors contributed equally: Fei Shen, Yajuan Qin, Rui Wang. ✉e-mail: weijianhua@baafs.net.cn; lei.li@pku.edu.cn; yangxz@sRNAworld.com

are spread over a greater distance than most other types of seeds. In addition, inulin-type fructans, instead of starches, are the primary reserve carbohydrates in the Asteraceae, and they have potential functions to increase their ability to adapt to environmental challenges[6–8]. However, progressively understanding the explosive diversifications and adaptability of Asteraceae still remains a strong challenge to biologists.

The origin and early evolution of the Asteraceae is inconclusive and mysterious. The recent phylogenetic studies of Asteraceae using new fossil evidence and a broader sampling of the family placed its origin sometime in the late Cretaceous period (69–89 million years ago [MYA])[2,5,9]. The family was considered to be relatively young until recently (40–50 MYA) based on its existing fossil record, and this time frame is consistent with that provided by molecular clocks[2,10,11]. A paleopolyploidization event was proposed and shared by subfamilies near the crown node of Asteraceae, which was considered to be a whole-genome triplication by recent genomic analyses[10,12–14]. In addition, frequent potential ancient whole-genome duplications (WGDs) within several tribes were estimated and predicted to be a force that drives evolution and increases biodiversity considering that polyploidizations duplicate all genes simultaneously and provide abundant genetic materials for evolutionary processes, such as neofunctionalization, subfunctionalization, and gene conservation owing to dosage effects[13,15]. Insights into polyploidization enable the genetic base of specific traits of Asteraceae to become feasible by delving into the high-quality genomes utilizing cutting-edge sequencing technologies.

Nitrogen (N), along with carbon (C), is a primary constituent of the nucleotides and proteins that are essential for life, but its availability is often a limiting factor for plant growth in natural ecosystems. Cellular N and C metabolism in plants are finely coordinated by sensor or regulatory genes to sustain optimal growth and development[16]. For example, the PII proteins act as reporters of the C metabolic state of the cell by interdependently binding ATP/ADP and 2-oxoglutarate (2-OG)[17]. Furthermore, the levels of cellular glutamine in plants are additionally sensed via PII signaling[16–18]. The mode of function of PII is conserved in the three domains of life under the control of PII-target protein interactions via the binding of effector molecules. Ecologically successful taxa evolve successful and efficient N assimilation systems to survive in severe habitats or compete for nourishment. Approximately 90% of the species within the family Leguminosae can fix atmospheric N through a symbiotic association with soil bacteria and have become widespread through the most spectacular radiations[19]. Orchids are one of the very few flowering plant lineages that have been able to successfully colonize epiphytic or lithophytic niches, clinging to trees or rocks and growing in dry conditions using crassulacean acid metabolism[20]. We reasonably hypothesize that there could be unusual factors in the nutrient absorption system of Asteraceae.

In this work, we generate two high-quality genome assemblies of stem lettuce (*Lactuca sativa* var. *angustana*), a representative economic crop of Asteraceae, and *Scaevola taccada*, a representative plant of the Goodeniaceae family that is the sister lineage to Calyceraceae and Asteraceae (Fig. 1a, b). A comparative genomics analysis of all the

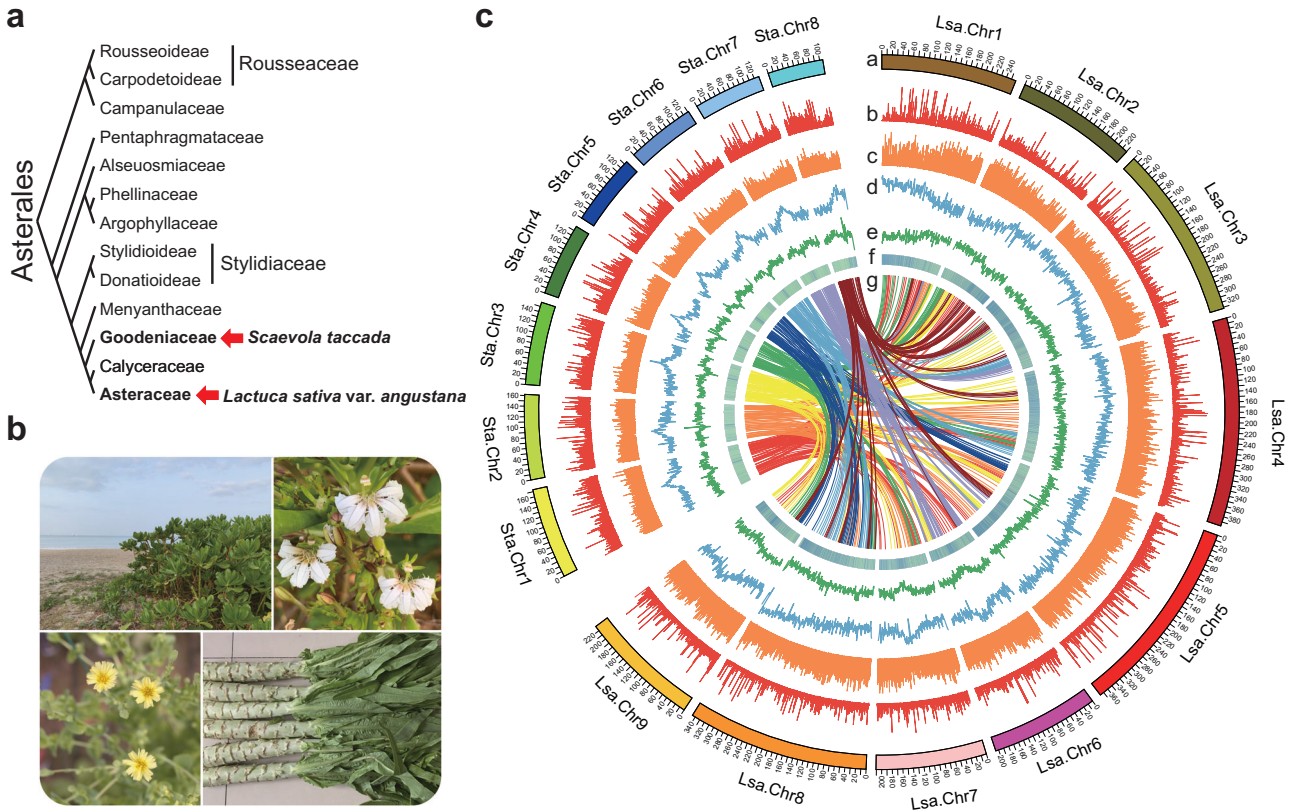

**Fig. 1 | Phylogenic relationship and genome features of stem lettuce (*Lactuca sativa* var. *angustana*) and *Scaevola taccada*. a** Phylogenic relationship between stem lettuce and *Sc. taccada*. Red arrows indicate their respective families, Asteraceae and Goodeniaceae. The tree was adapted from Angiosperm Phylogeny (http://www.mobot.org/MOBOT/research/-APweb/). **b** Habitat, morphology, and flowers of stem lettuce and *Sc. taccada*. Typical habitat of *Sc. taccada* on a tropical beach in Sanya, Hainan Province, China (top left); Flowers of *Sc. taccada* (top right); Morphology of stem lettuce as a commodity in a supermarket (bottom right); Flowers of stem lettuce (bottom left). **c** Genome features of stem lettuce and *Sc. taccada*. The tracks from outer to inner circles indicate I, chromosome karyotypes; II, gene density in 1-Mb windows; III, gene expression levels (averaged FPKM in 1-Mb windows); IV, density of LTR-RTs (1-Mb windows); V, DNA transposon density (1-Mb windows); VI, GC content; VII, synteny between the two genomes.

29 taxa, including seven Asteraceae species and 22 species, that are representatives of different evolutionary clades of terrestrial plants is performed. It uncovers the history of genomic evolution, genomic architecture and gene functional differentiation of Asteraceae, which could possibly lead to its uniqueness and diversity. Strikingly, the conserved regulatory gene that maintained the balance of N/C assimilation, *PII*, is lost across the Asteraceae family during its genomic transition. A unique N-C balance system has been proposed and covers but is not limited to the absorption of N and fatty acid synthesis and provides a solid molecular basis for the adaptability of Asteraceae.

## Results

### Two high-quality de novo genome assemblies

The de novo genome assembly for stem lettuce was based on a set of 105-fold-coverage single-nucleotide real-time (SMRT) sequencing, 119-fold-coverage optical mapping, 108-fold-coverage chromosome conformation capture (Hi-C) sequencing, and 158-fold-coverage Illumina reads (Supplementary Table 1 and Supplementary Notes 2 and 3). The N50 sizes of the contigs, scaffolds with optical mapping, and scaffolds further analyzed with Hi-C, were 4.95 Mb, 186.5 Mb, and 332.3 Mb, respectively (Supplementary Tables 2 and 3). The final assembled genome, namely SL1.0, is 2589.7 Mb, including nine pseudochromosomes and the complete genomes of chloroplasts and mitochondria (Fig. 1c, Supplementary Tables 2 and 3, Supplementary Note 4, and Supplementary Figs. 2–5).

The RNA-Seq datasets from different tissues produced by this study and all the expressed sequence tags (ESTs) of lettuce from NCBI were used to predict the genes and annotate the genome (Supplementary Note 5 and Supplementary Fig. 6). A total of 40,341 high-confidence protein-coding genes and 5453 non-coding RNAs were identified, and their functions were annotated by searching against six publicly available databases (Supplementary Tables 4 and 5). Finally, together with annotation against the Gene Ontology (GO) and Kyoto Encyclopedia of Genes and Genomes (KEGG), more than 85% of the genes were annotated (Supplementary Table 4).

Different means were utilized to assess the genome quality (Supplementary Note 6 and Supplementary Figs. 7–9). First, a pairwise alignment between SL1.0 and the genome of another lettuce cultivar (*La. sativa* var. *capitata*) was conducted, and in addition to its strong collinearity and consistency, SL1.0 was much more complete (Supplementary Fig. 8). Secondly, 98.37% of all the Illumina reads and 95.32% of ESTs, respectively, could be properly re-aligned to the SL1.0 assembly (Supplementary Tables 6–8), and its base accuracy, designated the quality value (QV), was estimated to be at least 42.38, which was better than the two favorable published mammalian genomes (QV35 for gorilla and QV34.5 for goat)[21,22]. Third, the genome covered 95.9% of the complete Benchmarking Universal Single-Copy Orthologs (BUSCO), and ~92.3% of them were expressed in at least one tissue (Supplementary Table 9 and Supplementary Fig. 7). A notable feature of SL1.0 is the high portion of repetitive elements that comprised more than 87% (Supplementary Table 10). After a careful examination of the long terminal repeats (LTRs), the LTR assembly index (LAI) of SL1.0 is as high as 18.13, comparable to the quality of model plants, such as *Arabidopsis thaliana*, rice, and maize (Supplementary Fig. 9)[23]. Moreover, SL1.0 captured five long stretches of telomeric sequences at both ends of the five chromosomes, with repeat numbers that ranged from 294 to 1073 (Supplementary Table 11). The combination previously described proved that the genome of stem lettuce is a high quality genome in Asteraceae.

Similarly, the de novo genome assembly for *Sc. taccada* was based on a set of 102-fold coverage SMRT reads, 102-fold coverage of Hi-C reads, and 105-fold coverage of Illumina reads (Supplementary Table 12, Supplementary Notes 7 and 8, and Supplementary Figs. 10–13). We succeeded in generating the assembly of *Sc. taccada*, namely ST1.0, with a size of 1159 Mb, including eight chromosome-scale scaffolds where contig N50 was as high as 9.6 Mb, and with complete genomes of the chloroplasts and mitochondria (Fig. 1c, Supplementary Tables 13 and 14, Supplementary Note 9, and Supplementary Figs. 11–13). The repeat sequences covered ~952 Mbp (over 80% of the assembly), and the LTR retrotransposons (LTR-RTs) occupied nearly 82% of all the repeats (Supplementary Table 15 and Supplementary Note 10). When the RNA-Seq datasets from four different tissues were utilized to annotate ST1.0, 25,328 protein-coding genes and 4219 non-coding RNAs were obtained with detailed annotation information (Supplementary Tables 16 and 17 and Supplementary Note 10). In addition, the repeat sequences of the genome were well assembled with the LAI as high as 12.01. The BUSCO estimated completeness of ST1.0 reached 94.2% (Supplementary Table 18 and Supplementary Fig. 14). All the other assessments that were determined using methods identical to those of SL1.0 indicate a high quality and completeness of ST1.0 (Supplementary Tables 18 and 19).

### Cretaceous origin and ancestral whole-genome triplication event

To investigate the origin and genome evolution of Asteraceae, we used the genomes of *Sc. taccada* and 21 representatives of different phylogenetic branches of land plants and all seven sequenced Asteraceae, including lettuce (Fig. 2a, Supplementary Figs. 15 and 16, and Supplementary Note 11). We traced the lineage differentiation time of *Sc. taccada* and Asteraceae to approximately 78–82 million years ago (MYA) in the late Cretaceous period (Fig. 2a), which was consistent with findings from recent curation of pollen grain fossil records and molecular analysis based on dispersed genome and transcriptome sequences[2,5].

We calculated the number of synonymous substitutions per synonymous site (*Ks*) of all paralogs. The *Ks* distribution revealed different polyploidization events experienced by the Asteraceae species, including the ancestral whole-genome triplication of the Eudicots (WGT-γ)[24], whole-genome triplication that share by the Asteraceae species (WGT-1)[12,14], and a lineage-specific whole-genome duplication in the sunflower genome (WGD-2)[10] (Fig. 2b and Supplementary Note 12). *Sc. taccada* shared a clear *Ks* peak with coffee (*Coffea arabica*), suggesting that *Sc. taccada* and coffee experienced only the WGT-γ event, an ancient WGT event that occurred approximately 122–164 MYA[10,24], which was consistent with previous results from *Scaevola aemula* transcriptome[25]. The similar *Ks* distribution profiles of paralogs between *Sc. taccada* versus lettuce and lettuce versus lettuce indicated that the WGT event occurred shortly after the cladogenesis between Asteraceae and Goodeniaceae (Fig. 2b and Supplementary Fig. 17a). WGT-1 was the only WGT event that happened in the Asteraceae ancestors after WGT-γ. The intergenomic comparison and synteny analyses also supported this observation (Fig. 2c and Supplementary Note 12). In addition, these analyses suggest that the WGT-1 event took place near the time of formation of Asteraceae (Fig. 2a).

We also investigated the triplication-retained regions (TRRs) after the WGT-1 event using the *Sc. taccada* genome as the reference (Supplementary Table 20, Supplementary Note 13, and Supplementary Fig. 18). We detected key homologous genes responsible for essential biological processes (e.g., flowering, cell wall biosynthesis/metabolism, fatty acid biosynthesis) in the TRRs. Further analysis of TRR-enriched genes revealed that genes related to the cell wall, protein phosphorylation, fatty acid biosynthesis, and cell membrane were selectively retained (Supplementary Fig. 17b, Supplementary Data 1–5, and Supplementary Note 13). All of these functions are closely related to stress responses and environmental adaptation, for example, pectin methylesterases (PMEs) can facilitate cell wall modification. Genes encoding delta-9 acyl-lipid desaturases (ADSs) and MIKC-MADS transcription factors exhibited

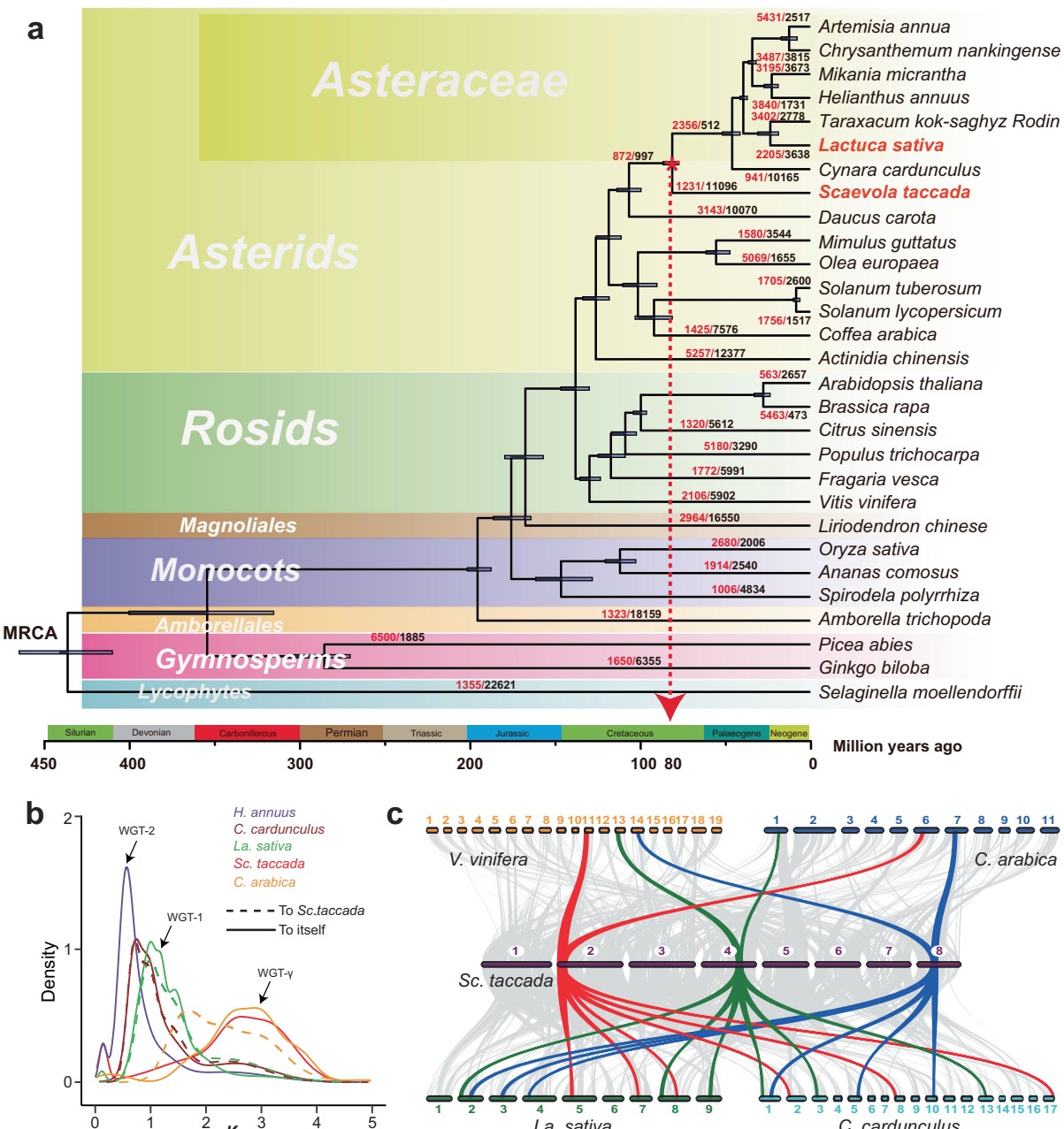

**Fig. 2 | Phylogenetic relationship of Asteraceae with other plants and whole-genome triplication events. a** Phylogeny and timescale of 29 representative terrestrial plant species. Gray bars at nodes represent 95% confidence intervals of the estimated divergence dates, while red arrow and star indicate the origin of Asteraceae around 80 million years ago (MYA) in the late Cretaceous. Numbers at each branch or node represent expansions (red) and contractions (black) of orthologous groups. MRCA: most recent common ancestor. **b** Density distribution of estimated synonymous substitution rate ($K_s$) of syntelog pairs for intragenomic comparisons (*Sc. taccada*, lettuce (*La. sativa*), sunflower (*Helianthus annuus*), artichoke (*Cynara cardunculus*), and coffee (*C. arabica*)) and for *Sc. taccad*a versus coffee, lettuce, and artichoke. **c** Macrosynteny visualization of the genomes of grapevine (*Vitis vinifera*), coffee, *Sc. taccada*, lettuce, and artichoke showing triplication events in Asteraceae. Three examples of syntelogs are colored in red, green, and blue, where one copy is in grapevine, coffee, and *Sc. taccada*, and three copies are in lettuce and artichoke. Numbers indicate chromosomes. Source data are provided as a Source data file.

a similar retention pattern (Supplementary Figs. 17c, 19, and 20 and Supplementary Note 13). The distinct genomic composition of TRRs with significantly higher genic regions and lower repetitive sequences compared to that of the whole genome indicates that these regions were preferentially selected (Supplementary Fig. 17d and Supplementary Note 13). An enrichment and depletion analysis of the repeat element families in these TRRs indicated that the decrease of the repeat elements was primarily caused by the deletion of LTR-RTs (Supplementary Fig. 21 and Supplementary Note 14), which have substantially expanded and dominate

Asteraceae genomes (Supplementary Note 14, Supplementary Data 6–10, Supplementary Table 21, and Supplementary Figs. 22–35).

## The evolution of gene families
Next, we analyzed gene family evolution based on the phylogenetic tree and orthologous groups in the ancestral node of Asteraceae and the ancestral node of the Asteraceae and Goodeniaceae families (Fig. 2a, Supplementary Note 15, and Supplementary Fig. 36). The rapidly evolving gene families were involved in a wide range of

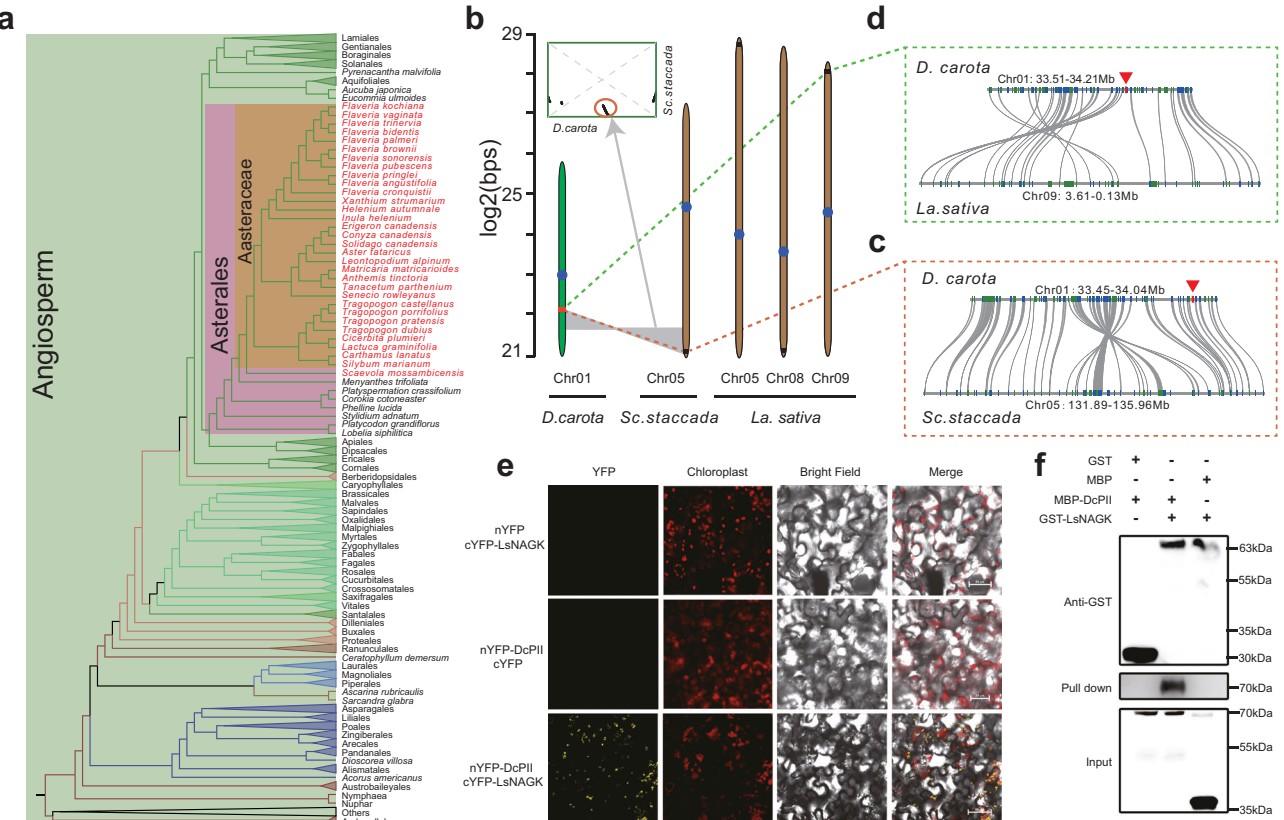

**Fig. 3 | PII is lost in Asteraceae. a** Phylogenetic distribution of *PII* in 1090 green plants from the One Thousand Plant Transcriptomes Initiative (OTPTI) collection. A simplified phylogenetic tree is displayed on the left and species or other taxonomic names are listed on the right. Clades without *PII* are labeled in red font. **b** Colinearity analysis of the *PII*-containing region in carrot (*D. carota*), *Sc. taccada*, and lettuce. *PII*-containing chromosome 1 in carrot and the corresponding assembled chromosomes in *Sc. taccada* and lettuce are shown vertically. Red and black lines on the chromosomes indicate the existence and loss of *PII*, respectively. The gray shade and genome syntelog in the inset show the chromosomal rearrangement between carrot and *Sc. taccada*. **c**, **d** Microsynteny visualization of the colinear chromosome regions between carrot and *Sc. taccada* (**c**) and between carrot and lettuce (**d**). The position of *PII* in carrot is marked with red arrows. **e** Bimolecular fluorescence complementation (BiFC) analysis showing the interaction between LsNAGK and PII (DcPII) in vivo. **f** GST pull-down assays showing the interaction between LsNAGK and PII (DcPII) in vitro. Three times each experiment was repeated independently with similar results. Source data are provided as a Source data file.

biological processes, particularly reproduction, response to stimuli, immunity, and nutrient reservoirs (Supplementary Fig. 37). In parallel, we analyzed the functional domains of these genes and identified 107 Interpro entries that were enriched in at least two Asteraceae species ($P < 0.05$) (Supplementary Data 11 and Supplementary Note 16). Notably, 18 entries were related to (retro)transposon domains, and several types of zinc finger domains, including Zinc finger CCHC-type, BED-type, PMZ-type, TTF-type, SWIM-type, and GRF-type were also identified (Supplementary Data 11). Another enriched group was related to fatty acid biosynthesis, such as acyl-CoA desaturases, beta-ketoacyl synthases, and fatty acid desaturases (FADs) (Supplementary Data 11). We also identified 114 orthologous gene groups in four categories as lineage-specific in Asteraceae, including transcription factor genes such as *bHLH*, *MYB*, *Znf*, SPL, and MADS-box, a subclade of the FERONIA receptor kinase family, genes for key secondary metabolism in Asteraceae (e.g., inulin and alkaloids biosynthesis), and cell remodeling (Supplementary Note 17, Supplementary Tables 22 and 23, Supplementary Data 12 and 13, and Supplementary Figs. 38–48).

**The loss of PII and its influences**

We also investigated genes that were absent in the seven sequenced Asteraceae species (Supplementary Table 24 and Supplementary Note 18). The most conspicuous absence in all seven Asteraceae and *Sc. taccada* was *PII*. *PII* occurs widely in all three domains of life, and has a pivotal role in sensing and regulating N-C signals[18]. We used the transcriptome data from the One Thousand Plant Transcriptomes (1KP)

Initiative[26], including 39 Asteraceae species and fan flower (*Scaevola mossambicensis*), another Goodeniaceae species, and confirmed the loss of *PII* in the Asteraceae and Goodeniaceae (Fig. 3a, Supplementary Note 19, Supplementary Data 14, and Supplementary Table 25), indicating that *PII* was lost in the ancestor of Asteraceae and Goodeniaceae.

To trace how *PII* was lost, we selected another close outgroup species, carrot (*Daucus carota*), as the reference, and conducted pairwise syntenic analyses with *Sc. taccada* and other Asteraceae species (Fig. 3b and Supplementary Note 19). In contrast to carrot, in which the *PII*-containing syntenic block maps to the middle of the lower arm of chromosome 1, this syntenic block was located along the telomere regions in the Asteraceae and Goodeniaceae genomes (Fig. 3b). Pairwise synteny indicated an extensive chromosomal rearrangement between carrot (Chr 01) and *Sc. taccada* (Chr 05), and *PII* was located on the edge of the rearranged area (Fig. 3b). Furthermore, we detected a micro-inversion (involving 20−30 genes) between carrot and *Sc. taccada* (Fig. 3c) and between carrot and lettuce (Fig. 3d). *PII* was located on the border of the inverted region (Fig. 3c, d). Taking these results together, we propose that a chromosomal rearrangement followed by a micro-inversion occurred in the ancestor of Goodeniaceae and Asteraceae, which led to the loss of *PII*.

PII is a chloroplast-localized N sensor that activates the *N*-acetyl-L-glutamate kinase (NAGK) complex to promote N assimilation[16,17]. PII also forms a complex with the biotin carboxyl carrier protein (BCCP) subunit of acetyl-CoA carboxylase (ACCase, which catalyzes the first

step in fatty acid biosynthesis) to inhibit ACCase activity[18,27]. Moreover, PII may participate in the negative regulation of N uptake and prevent overexcess of nitrite uptake in land plants[28,29]. To evaluate the influence of *PII* absence in Asteraceae, we generated transgenic lettuce plants expressing an exogenous *PII* gene from carrot (*Da. carota*), tomato (*Solanum lycopersicum*) and Arabidopsis (*A. thaliana*), respectively. The PIIs from carrot/tomato represent the canonical PIIs with plant-specific glutamine-binding sites that were partly deleted in that of Arabidopsis (Supplementary Note 20 and Supplementary Fig. 49). Consistent with previous reports[30], AtPII localized to the chloroplast in *AtPII*-expressing transgenic lettuce (Fig. 3e and Supplementary Figs. 50 and 51). The in vivo assays (Bimolecular Fluorescence Complementation-BiFC) and in vitro tests (Pull-down) strongly support the interactions between AtPII, DcPII, SlPII and NAGK in lettuce (Fig. 3e, f, Supplementary Fig. 52, and Supplementary Note 20). The nitrate N content and ACCase activity were significantly lower in the transgenic plants (Supplementary Fig. 52 and Supplementary Note 20). Compared to wild-type lettuce, the total content of free amino acids, especially glutamic acid, glutamine and arginine, was significantly increased in the *DcPII*- and *SlPII*-expressing transgenic plants, while reduced in *AtPII*-expressing lines (Supplementary Fig. 52 and Supplementary Note 20). Although the specific reasons are deserved to explore, differences in glutamine-binding sites of PII proteins were highly suspected to have an effect (Supplementary Note 21). These results indicate that exogenous PII could disturb the original N-C balance in lettuce.

### Preferential retention and divergence of N absorption genes

The close-to-universal presence of PII in almost all domains of life indicates that its absence in the Asteraceae should result in a major reprogramming of the N-C balance system. Therefore, we examined the different genomes for the presence of genes involved in N absorption and metabolism. Asteraceae had significantly more members of the *Nitrate transporter 2* (*NRT2*) and *NRT3* gene families, which encode dual-component transporters involved in high-affinity nitrate assimilation[31], compared to the other species, including *Sc. taccada* ($P < 0.01$; Fig. 4a, f). This observation was consistent with the rapid expansion of N reservoir orthologs in the crown node of Asteraceae (Supplementary Fig. 37 and Supplementary Note 22).

Colinearity analysis between lettuce and *Sc. taccada* revealed that the expansion of the *NRT2* and *NRT3* families was initiated by the WGT event and mainly arose by tandem duplications (Fig. 4c, h). The *NRT2s* of Asteraceae primarily grouped in clades I and II of a reconstructed phylogenetic tree (Fig. 4d and Supplementary Fig. 53) with preferential expression in the root (Fig. 4e and Supplementary Fig. 54). These genes clustered together with key members of high-affinity nitrate transporters in Arabidopsis, including genes mainly functioning in roots, such as *NRT2.1* (At1g08090), *NRT2.2* (At1g08100), and *NRT2.6* (At3g45060) (Supplementary Fig. 53 and Supplementary Note 22)[31]. Similarly, *NRT3* members in Asteraceae mainly grouped into two clades (II and III; Fig.4i) and were preferentially expressed in the root (Fig. 4j and Supplementary Figs. 55 and 56). The $Ka/Ks$ value of orthologous gene pairs and paralogous gene pairs of clade I *NRT2* and clade III *NRT3* members indicated that all the duplicated genes were subjected to purifying selection during evolution (Fig. 4b, g and Supplementary Note 22). These observations demonstrate an expansion of the high-affinity nitrate transporters in Asteraceae via WGT and subsequent tandem duplications.

### The reinforcement of fatty acid metabolism

Another strong genomic footprint of PII loss was that the genes associated with fatty acid metabolism in Asteraceae were enriched in TRRs, had rapidly expanded, and had numerous InterPro entries (Supplementary Data 1–5, Supplementary Fig. 17b and 57–59, and Supplementary Note 23). Accordingly, we analyzed the key genes in fatty acid

biosynthesis, i.e. *ADSs*, *FADs*, and *3-oxoacyl-[acyl-carrier-protein] synthases* (*KASs*). All three families were significantly expanded in Asteraceae compared to the other species ($P < 0.01$; Fig. 5a and Supplementary Figs. 60–63). For example, the *Sc. taccada* genome contained only three *FAD* genes, whereas those of Asteraceae species contained at least 14 *FADs* (Fig. 5b and Supplementary Note 23). We explored the mechanisms governing the expansion of these gene families. Similar to the *NRT* families, expansion of the *ADS*, *FAD*, and *KAS* families initially occurred via the WGT event and subsequently by tandem duplications (Fig. 5c, Supplementary Note 23, and Supplementary Figs. 60–65).

### Proposal of a unique N-C metabolism balance system

Genome sequencing and comparison revealed a unique scenario in Asteraceae evolution. Before the split between Asteraceae and Goodeniaceae, a large chromosomal inversion placed the *PII* locus near the telomere, which we hypothesize was subsequently lost due to a micro-inversion (Fig. 3). Compared to other higher plants and Goodeniaceae species, the Asteraceae evolved a stepwise upgrade of the N-C balance system, initially through the WGT that occurred approximately 78–82 MYA (Fig. 2a), and subsequently by tandem duplications. In the Asteraceae system, expansion of the high-affinity nitrate transporter genes potentially increased their ability to take up N (Fig. 6 and Supplementary Note 22), especially in N poor environments (Fig. 4). Moreover, the removal of PII inhibition would increase ACCase activity (Fig. 3) and provide more substrates for fatty acid biosynthesis (Fig. 6b and Supplementary Note 23, which would be fulfilled by the expansion of the fatty acid biosynthesis genes *FADs*, *KASs*, and *ADSs* (Fig. 5). Therefore, we propose that the Asteraceae evolved a unique N-C balance system following the loss of *PII*, resulting in enhanced N uptake capacity and fatty acid biosynthesis (Fig. 6), which may explain their high biodiversity and excellent adaptability.

## Discussion

Genomic novelty based on genome duplications contributes to speciation, adaptive radiation, and is consequently likely to enable organisms to utilize new ecological opportunities or to manage new environmental challenges[13,15,32]. Consistent with previous reports[2,9,12], we estimated that the early paleopolyploidization in Asteraceae happened near the speciation of Asteraceae and Goodeniaceae, tracing back to the late Cretaceous period (~80 MYA), which possibly provided genetic materials to manage a series of explosive radiations during the Eocene. Limited empirical evidence suggests that the commonly retained duplicate genes after paleopolyploidization in the critical stress-related pathways could be the key factors that partially improve adaptability. For example, genes that alter the cell wall in response to low temperature and darkness were commonly retained after WGDs when global cooling and darkness were the two primary stresses[33]. Investigating the duplicates of the MADS-box gene family in the core eudicots suggested that the WGT-γ event likely initiated the functional diversification of the developmental regulators of floral organs, favored the morphological innovation of flowers and potentially promoted the adaptive radiation of core eudicots[32]. In particular, using *Sc. taccada* as a reference enabled us to independently obtain retained genes after the paleopolyploidization in the Asteraceae species. Interestingly, we observed that the genes associated with cell wall biosynthesis, protein phosphatase, flowering and fat acid biosynthesis were simultaneously enriched in the TRRs of surveyed Asteraceae genomes ($P < 0.05$), and several vital families are related to adaptability, such as the MADS-box, *PMEs* and *ADSs* (Supplementary Fig. 17b). Therefore, these biased gene retention after WGDs is potentially related to the adaptability and speciation of Asteraceae.

In addition, sub/neo-functionalization of vital duplicate genes increase genetic diversity and possibly facilitate adaptive evolution in

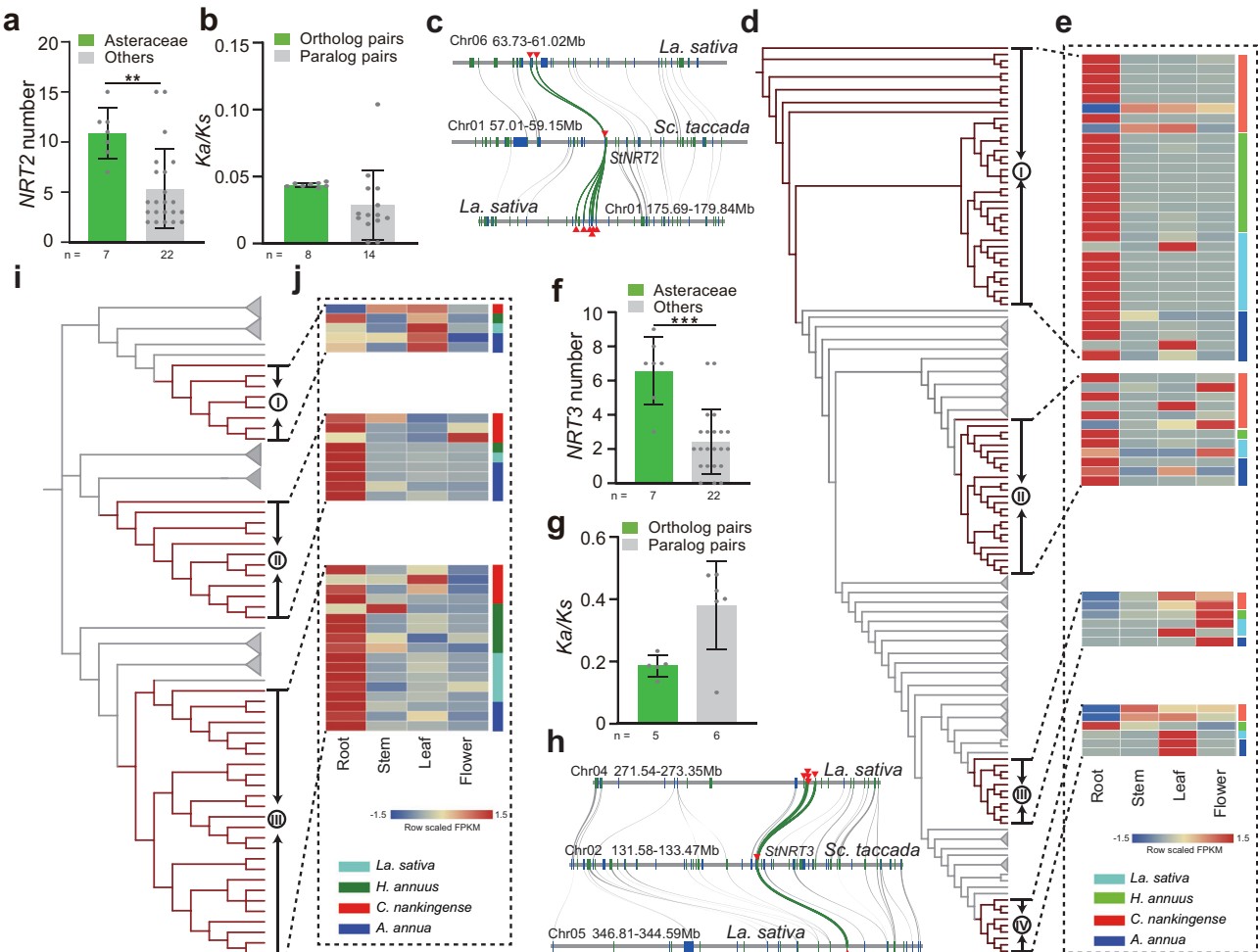

**Fig. 4 | Expansion and preferential retention of N uptake genes in Asteraceae.**
**a**, **f** Number of genes for the *Nitrate transporter 2* (*NRT2*) and *NRT3* families in Asteraceae and other terrestrial plant species. **\*\****P* < 0.01 and **\*\*\****P* < 0.001, as determined by two-tailed Student's *t* test. The number (*n*) of data point for each group was shown below. Data are presented as mean values +/−SD. **b**, **g** $K_a/K_s$ value of orthologous gene pairs (*Sc. taccada* versus *La. sativa*) and paralogous gene pairs (*La. sativa* versus *La. sativa*) of clade I *NRT2* (**d**) and clade III *NRT3* (**i**) members. **c**, **h** Microsynteny visualization of clade I *NRT2* (**d**) and clade III *NRT3* (**i**) syntelogs in

the *Sc. taccada* and *La. sativa* genomes. *NRT2* and *NRT3* orthologous pairs are highlighted with green lines. **d**, **i** Simplified phylogenetic tree of NRT2 (**d**) and NRT3 (**i**) in Asteraceae species and other species. The genes from the Asteraceae species are marked by dark red, while clades from other species are collapsed. **e**, **j** *NRT2* (**e**) and *NRT3* (**j**) expression in different tissues of four representative Asteraceae species. The FPKM values of genes in different tissues are scaled by rows to emphasize the tissue with the highest expression. Source data are provided as a Source data file.

Asteraceae. Such genes include essential regulators and defense-related functional genes (Supplementary Note 17). Impressively, the signs of dramatic amplification and differentiation of the *FERONA* gene family led to a mass of lineage-specific genes, accompanied with the emergence of Asteraceae-specific miRNAs (Supplementary Note 17). Moreover, the gene divergence of the biosynthetic pathways for alkaloids and inulin potentially determined the genetic basis for these characteristics in Asteraceae (Supplementary Note 17). As the most abundant component of the genomes, (retro)transposons, function by their insertions into genes or promoter regions, are additional key drivers that prompt the divergence of genes and genomes in plants. For example, the transposase-derived proteins FHY3/FAR1 regulate chlorophyll biosynthesis in *A. thaliana*[34]. Here, we observed that (retro)transposon-associated genes were significantly higher in the Asteraceae, including several key genes, such as *RVT-Znf* and *RT_RNaseH_2. BDR4* (Supplementary Fig. 22f). In addition, the genes with potentially regulatory DNA elements derived from (retro)transposable elements were involved a wide range of biological functions, such as *BDR4* (Supplementary Note 14 and Supplementary Fig. 22f). Our observations were consistent with other previous studies that reflected the important roles of repetitive sequences in diversifying

the Asteraceae genomes[35,36]. All of the data discussed above provide evidence for drivers and the impacts of genomic and gene dynamics on species radiation and conservative/innovation characteristics of the Asteraceae family.

Powerful nitrogen uptake and absorption systems are essential for plants to survive in nutrient-limited environments and particularly prevalent in Asteraceae. As a representative of rapid growth Asteraceae plant, *M. micrantha*, its metabolites can increase the availability of nitrogen by enriching the microbes that participate in nitrogen cycling pathways[37]. Here, we identified that *PII*, playing a key role in sensing nitrogen and carbon signals in all domains of life[18] has been lost in the Asteraceae plants. When expressing exogenous *PII* into lettuce, PII proteins can still interact with NAGK and inhibit the ACCase activity, which in turn affects many physiological and metabolic pathways such as disturbing the synthesis of amino acids and N uptake (nitrate). These changes indicate that Asteraceae, represented by lettuce, has evolved a unique N-C balance system. Given that the PII sensing system is beneficial to maintain metabolic homeostasis under fluctuating nitrogen supply, it is possible that the loss of PII in the Asteraceae ancestors was not disadvantageous. Because of the increase of *NRT2/3* genes via WGT and tandem duplication, the *NRT2/3*

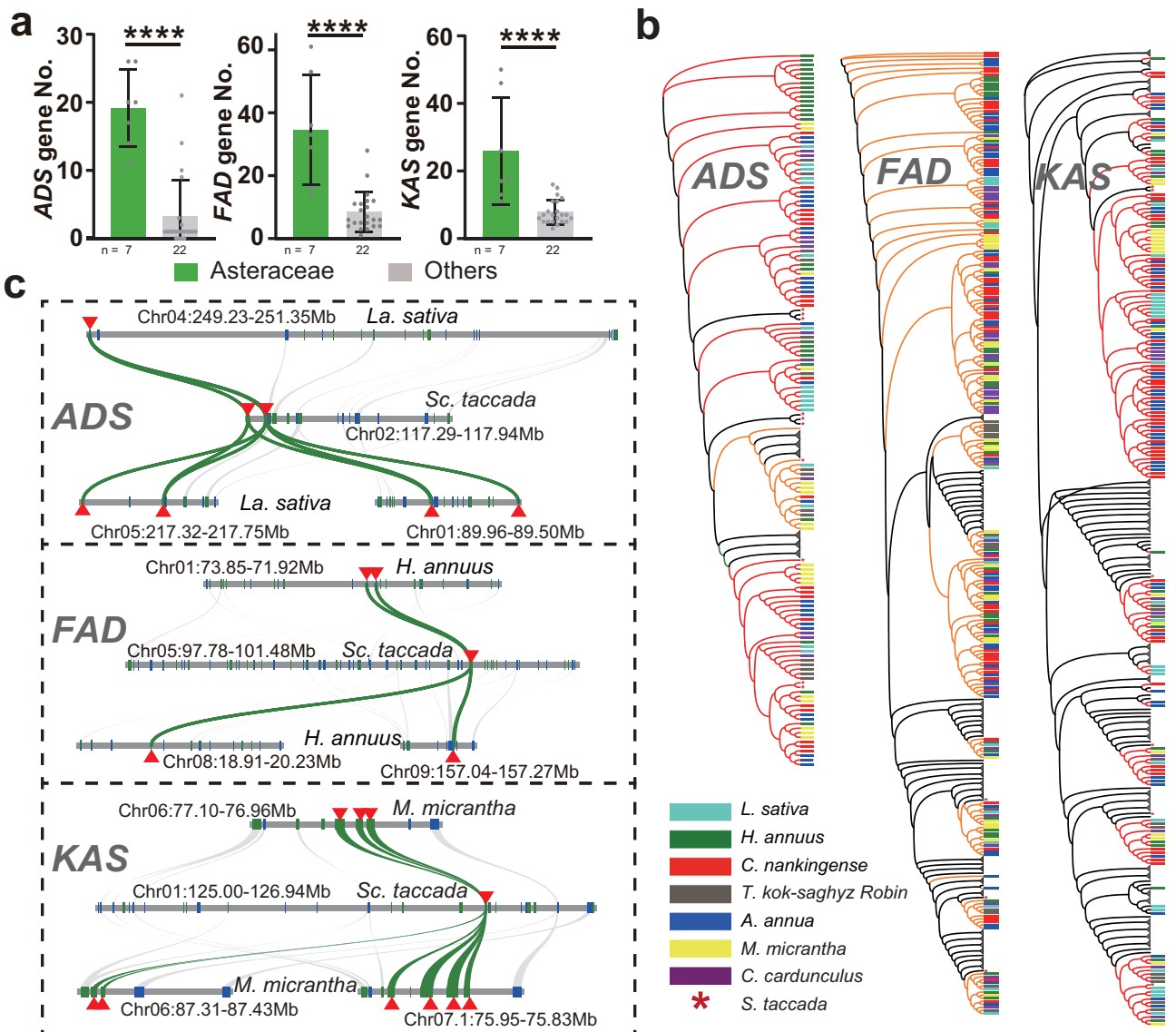

**Fig. 5 | Expansion of the *ADS*, *FAD*, and *KAS* gene families in Asteraceae.**
**a** Number of genes for *ADSs*, *FADs*, and *KASs* in the seven Asteraceae species and 22 other land plants. ****$P < 0.0001$, as determined by two-tailed Student's *t* test. The number ($n$) of data point for each group was shown below the *x*-axes. Data are presented as mean values +/−SD. **b** Simplified phylogenetic trees of ADSs, FADs, and KASs in the seven Asteraceae species, *Sc. taccada*, and 21 representative land plants. The genes from the Asteraceae species and *Sc. taccada* are marked with different colors and stars as indicated, and clades from other species are collapsed. **c** Microsynteny visualization of typical *ADSs*, *FADs*, and *KASs* gene expansion patterns in the representative Asteraceae species and *Sc. taccada*. Source data are provided as a Source data file.

genes (as binary), so called high-affinity nitrate transporters, could absorb nitrate from a low nitrate concentration environment and provides a relatively abundant nitrogen supply, resulting in balancing the disadvantages of PII loss in Asteraceae. Under this situation, Asteraceae species further developed compensatory systems to offset the loss of PII, leading to their superior fitness and successive radiation.

The altered genetic basis for the N-C balance system definitively has a multifaceted influence on the plant physiology of Asteraceae and finally affects the ability to adapt to the environment. The gene expansion of high affinity nitrate transporters objectively increased the ability of plants to absorb nitrogen and thus, their adaptability. For example, glutamate signaling activates plant responses and adaptation to environmental stress[38], in addition to seed germination[39], root architecture[40], pollen germination and pollen tube growth[41,42]. The use of glutamate as a signaling molecule is also involved in the response and adaptation to salt, cold, heat, drought, pathogen, and wound stress[38,43,44]. PII regulates fatty acid synthesis in chloroplasts by interacting with the ACCase complex and inhibits its activity. In Asteraceae, relieving the inhibition of ACCase provides more possible substrates for the biosynthesis of fatty acids (FA) and UFA (Fig. 6). Plants have developed elaborate strategies with UFAs that have emerged as a general defender to avoid adverse effects[45,46].

Given that the loss of PII in Asteraceae might be occurred in the ancestor of Goodeniaceae and Asteraceae (~80 MYA), the unique N-C balance system in Asteraceae evolved for a long history, potentially resulting in complicated changes when compared to other plants with PII. More studies to fully investigate the physiological and metabolic adaptation machinery based on the unique N-C balance system in Asteraceae are further needed. Limited by geographical distribution and availability of genomic information, we were not able to include the Calyceraceae that is the other sister lineage to Asteraceae and shared WGT-1[25]. The adequate sampling of the genomes of

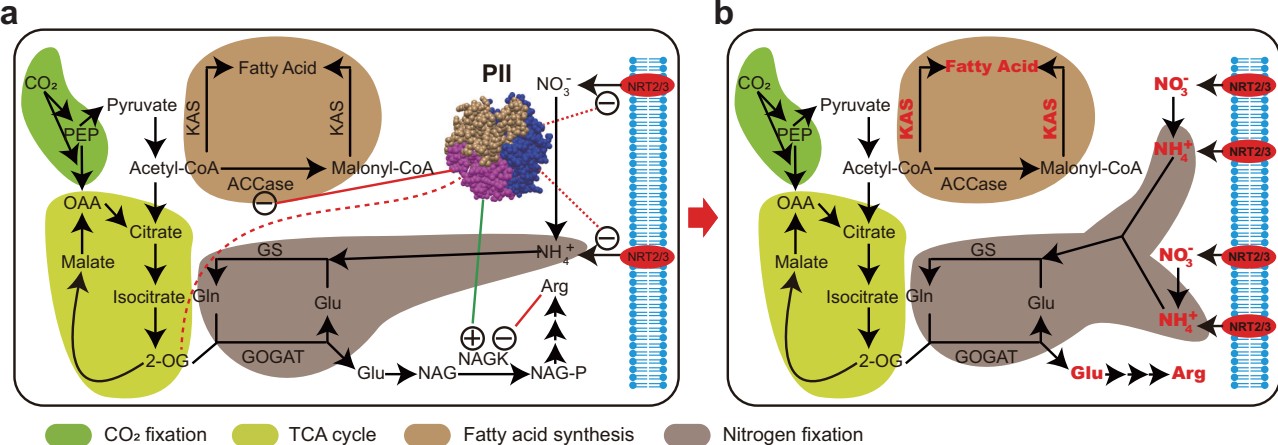

**Fig. 6 | Comparison of the N-C balance systems in Asteraceae and other species.** **a** Canonical N-C balance system in species containing PII. The structure of trimeric PII was downloaded from Protein Data Bank (https://www.ncbi.nlm.nih.gov/Structure/pdb/2O67). **b** The deduced N-C balance system in Asteraceae lacking *PII*. The strengthened uptake of N, and N and C assimilation processes are shown in bold red font. The regulation processes were indicated by lines, while the dotted ones represented the biological process that has not been directly confirmed but highly speculative.

Calyceraceae and basal subfamilies of Asteraceae (e.g., Barnadesioideae, Famatnanthoideae, and Stifftioideae) will help us fully reveal unique N-C balance system in Asteraceae and its evolutionary history in the future. Based on the detailed understanding of this unique N-C balance system, strategies by reconstructing the N-C system might be designed for crop improvement especially in meeting the global climate challenges.

## Methods

### Plant materials and sequencing

Stem lettuce (*Lactuca sativa* var. *angustana*) was planted in the greenhouse of the Beijing Academy of Agriculture and Forestry Sciences (BAAFS), Beijing (39°94´N and 116°28´E) in the spring of 2018, and *Scaevola taccada* seedlings were collected from Haikou, Hainan Province (20°03´N and 110°12´E) and planted in an artificial climate chamber at BAAFS in the summer of 2019.

Samples for de novo assembly were collected from a single lettuce plant and a single *Sc. taccada* plant. For single-molecule real-time (SMRT) sequencing, genomic DNA was isolated using a Blood & Cell Culture DNA Midi Kit (QIAGEN Inc., Valencia, CA, USA). The 20-kb libraries were constructed using SMRTbell Template Prep Kits (Pacific Biosciences, Menlo Park, CA, USA), and sequenced on a PacBio RS II instrument (Pacific Biosciences) using the P6-C4 sequencing reagent. For HiSeq analyses, DNA was isolated using Plant Genomic DNA kits (Tiangen, Beijing, China). Libraries with 450-bp inserts were prepared and sequenced on an Illumina HiSeq X platform (San Diego, CA, USA) for 150-bp paired-end reads. Young leaves were used for optical mapping by isolating high-molecular-weight DNA, which was then labeled using the direct labeling enzyme DLE-1 after isolation. The labeled DNA samples were imaged using a BioNano Irys system (Bionano Genomics, San Diego, CA, USA), and only molecules >150 kb were used for further analysis. For the Hi-C library, leaves were fixed in 1% (v/v) formaldehyde and the crosslinking reaction was terminated by adding glycine. Then, the leaf sections were removed from the mixture, rinsed with ddH₂O, and ground to a fine powder in liquid nitrogen to isolate cross-linked DNA. The isolated cross-linked DNA was purified, digested with *MboI* enzymes, and tagged with biotin. The biotin-tagged DNA fragments were captured and PCR enriched to construct the Hi-C library[47]. The Hi-C library was sequenced on an Illumina HiSeq X platform as 150-bp paired-end reads. Leaf, flower, root, and stem samples were collected separately for transcriptome deep sequencing (RNA-Seq). Total RNA was isolated using an RNAprep Pure Plant Kit (Tiangen). RNA-seq library construction was performed following the manufacturer's standard protocol (Illumina) and sequenced on an Illumina HiSeq X platform.

### Genome survey and assembly

Jellyfish software (v2.0)[48] was used to calculate the *k*-mer frequency (*k*-mer length 21), then Genomescope (v1.0.0)[49] was used to estimate genome heterozygosity, repeat content, and genome size from the sequence reads.

Tender leaves were collected from the sequenced *Sc. taccada* plant and analyzed using a flow cytometer. Black cottonwood (*Populus trichocarpa*) ($2n = 2x = 38$) and tomato (*Solanum lycopersicum*) ($2n = 2x = 24$) samples were analyzed as genome size references. Over 5000 nuclei per sample were collected and detected using a CyFlow Space flow cytometer (Partec, Germany) equipped with a UV-LED source (emission at 365 nm) and a blue solid-state laser (455 nm). The data were analyzed using Flomax2.8 (Sysmex Partec, France), with a coefficient of variation <5%.

The assembly of the stem lettuce genome was performed in a stepwise fashion. First, Falcon (v0.4)[50] was used to obtain the initial contigs. Then, an initial polishing step was performed with Arrow (v2.2.3) using PacBio-only long reads, and then Pilon (v1.20)[51] was used to correct the sequencing errors in the contigs with accurate Illumina short reads. Bionano optical maps were assembled into consensus physical maps using BioNano Solve v3.0.1 (https://bionanogenomics.com/). To anchor the hybrid scaffolds into chromosomes, the Hi-C sequencing data were aligned into scaffolds by Juicer (v1.5)[52] and 3D-DNA (v201008)[53].

The PacBio reads of the *Sc. taccada* genome were corrected using the reads correction module of the CANU pipeline (v1.7.1)[54]. De novo assembly was conducted using WTDBG2 software (v2.5)[55] in the CCS mode. To anchor the hybrid scaffolds into chromosomes, the Hi-C sequencing data were aligned into scaffolds as described for stem lettuce.

### Assessment of the genome assembly

The quality of the genome assembly was evaluated at different levels. Illumina reads with high single-base accuracy were aligned using BWA (0.7.12-r1039)[56]. Properly mapped reads were calculated to reflect the correct degree of assembly. The base accuracy of the sequencing was determined by calculating the quality value (QV) of the assembly[21,22]. Briefly, single nucleotide polymorphism (SNP) calling was conducted based on the aligned reads using SAMtools (v1.4)[57], and the number of SNPs was counted with Phred-scaled >30 and coverage >3 (*n*).

Simultaneously, the number of genome positions with read coverage >3 was also calculated (*N*). Finally, the QV of the assembly was calculated as:

$$QV = Log10\left(\frac{n}{N}\right) \qquad (1)$$

The completeness of the gene landscape was estimated using expression data, including RNA-seq and expressed sequence tag (EST) data. Lettuce ESTs from GenBank were aligned to the assembled genomes using the BLAST-like alignment tool (v0.36) with default parameters[58]. The reads generated in this study from different lettuce and *Sc. taccada* tissues (roots, leaves, flowers, and stems) were aligned to the two genomes using Hisat2 (v2.1.2)[59]. BUSCO was calculated to assess the genome assembly and annotation completeness with single-copy orthologs[60].

Pairwise alignment between our lettuce SL1.0 genome and the previously published leafy lettuce genome (the Lsa_v1 genome)[14] was performed by Minimap2 (v2.18)[61] and visualized via Minidot (https://github.com/thackl/minidot). The assembly of repeat sequences was evaluated by the long terminal repeat assembly index (LAI) program[23], which evaluates the contiguity of an assembly using long terminal repeat retrotransposons (LTR-RTs). The LTR_retriever pipeline (v2.9.0)[62] was used to integrate the candidate LTR-RTs identified by LTR_FINDER (v1.0.7)[63] and LTRharvest (v1.5.9)[64]. The whole-genome LAI was then calculated based on the LTR-RT library generated by LTR_retriever.

## Repeat analysis and gene annotation
The two genomes were annotated with the same pipeline. For repeat sequences, a customized repeat library was constructed to include known and novel repeat families. Miniature inverted transposable elements (MITEs) of the two assemblies were searched using MITE-Hunter (v1.0)[65] with default parameters. LTR_retriever pipeline was then used to integrate the candidate LTR-RTs identified by LTR_FINDER and LTRharvest. An initial repeat masking of the genomes was performed with the repeat library derived by combining the identified MITEs and LTR-RTs. The repeat-masked genome was uploaded into RepeatModeler (v1.0.11)[66] to identify repeat families. Finally, all the repeat sequences identified were combined and searched against a plant protein database (Swiss-Prot, https://www.uniprot.org/) that excluded proteins encoded by transposons. Elements with significant similarity to plant genes were removed. RepeatMasker (v4.0.6)[66] was used to search for similar transposable elements (TEs) in the Repbase TE library and customized repeat library.

The protein-coding genes were predicted from the repeat-masked genome using MAKER-P (v2.31.10)[67], which integrates evidence from protein homology, transcripts, and ab initio predictions. The homology-based evidence was derived by aligning protein sequences from seven plant species (*Arabidopsis thaliana*, lettuce, sunflower [*Helianthus annuus*], artichoke [*Cynara cardunculus* var. *scolymus*], *Chrysanthemum nankingense*, *Artemisia annua*, and rice [*Oryza sativa*]) to each genome assembly.

The RNA-seq data derived from four different libraries was assembled de novo using Trinity (v2.8.2)[68]. ESTs extracted from the NCBI nucleotide and EST databases (https://www.ncbi.nlm.nih.gov/nucleotide/) were also used to predict genes. First, all transcript sequences were uploaded to the PASA pipeline (v2.4.1)[68] to conduct alignment assembly. Five thousand complete gene models and sequences were extracted to train the parameters for SNAP (v1.0) and Augustus software (v3.0.3)[69,70]. All data and predictions were then used to produce a consensus gene set. Finally, the PASA pipeline was used again to refine the obtained gene model.

To refine microRNA (miRNA) identification, reads were aligned to the repeat-masked genome using BWA[56] and miRNAs were identified using miRDeep2 (v0.1.3)[71]. The transfer RNA and ribosomal RNA genes were predicted using tRNAscan-SE package (v2.0.0)[72] and RNAmmer (v1.2) algorithms with default parameters[73], respectively. Other non-coding RNAs were identified using Infernal cmscan (v1.1.4)[74] by searching against the Rfam database (https://rfam.xfam.org/, release 13.0).

## Evolutionary analysis
OrthoFinder (v2.4.0)[75] was employed for inference of orthologous groups in the 29 selected species. The 29 genomes consisted of 7 species from Asteraceae, 8 from the Asterids order, 6 from the Rosids clades, 3 from monocot clades, and 5 ancient species (*Ginkgo biloba*, Norway spruce [*Picea abies*], *Selaginella moellendorffii*, *Amborella trichopoda*, and Chinese tulip tree [*Liriodendron chinense*]). Low-copy-number (LCN) genes were identified based on OrthoFinder results with the requirements: strictly single copy in *La. sativa*, *Sc. taccada*, *Se. moellendorffii*, *G. biloba*, and grapevine (*Vitis vinifera*), and single copy in at least 5 of the 24 selected species[76].

Two independent methods were used to reconstruct the species tree of the 29 selected species. Multiple alignments were conducted using MUSCLE (v3.8.31)[77]. Next, trimAL (v1.2)[78] was used to trim low-quality aligned regions with the option "-automated1". LCN gene trees were estimated from the remaining sites using RAxML (v.7.7.8)[79] with the JTT + G + I model for amino acid sequences. The best-fit model was selected using ModelFinder under the Bayesian information criterion[80]. Phylogenetic reconstruction was performed stepwise with a carefully selected set of 9784 genes using the coalescence method implemented in ASTRAL (v5.5.1)[81]. In addition, the multiple alignment sequences of the genes in the 389 LCN OrthoGroups (OGs) were concatenated and the species tree was reconstructed using RAxML.

The 389 LCN genes (185,822 sites) in each species were concatenated and the tree topology inferred from our coalescent-based analysis of the 9784 genes from 29 taxa was fixed. Then, Bayesian phylogenomic dating analysis of the selected genes in MCMCtree, part of the PAML package (v4.10.0)[82], and approximate likelihood calculation for the branch lengths were performed. Molecular dating was conducted using an auto-correlated model of among-lineage rate variation, the JC69 substitution model, and a uniform prior on the relative node times. Posterior distributions of node ages were estimated using Markov chain Monte Carlo sampling, with samples drawn every 200 steps over 10 million steps following a burn-in of 200,000 steps. The penalized likelihood method under a variable substitution rate using r8s (v1.8.1) was also implemented. Three fossil calibrations corresponding to the crown groups of angiosperms (~126 Mya), eudicots (~120 Mya), and monocots (~113 Mya) were implemented as minimum age constraints in our penalized likelihood dating analysis[83]. The best smoothing parameter value of the concatenated LCN genes was determined by performing cross-validations of a range of smooth parameters from 0.01 to 10,000 (algorithm = TN; crossv = yes; cvstart = 0; cvinc = 0.5; cvnum = 15). Finally, a relaxed molecular clock was calibrated via pairwise divergence time on the TIMETREE website (http://www.timetree.org/)[84] and the species divergence time was estimated using r8s (v1.8.1)[85]. CAFÉ software (v4.2.1)[86] was used to compute gene family evolution based on the phylogenetic tree and orthologous groups.

## Whole-genome duplication and synteny analysis
The *Sc. taccada*, lettuce, sunflower, artichoke, *Ch. nankingense*, and coffee (*Coffea arabica*) genomes were compared. Synteny comparisons were identified by MCscan (v1.3.6)[87] with default parameters to predict paralogs and orthologs. The sequence divergence of paralogs (within each genome) and orthologs (between *Sc. sericea* and another genome) was calculated based on synonymous (*Ks*) substitutions using

the maximum likelihood method implemented in codeml of the PAML package[88] under the F3x4 model.

## Analysis of triplication-retained regions

The triplication-retained regions (TRRs) were defined across the chromosome-scale genomes of the Asteraceae species lettuce, artichoke, sunflower, and *Mikania micrantha*. No other WGD events occurred in the lettuce and artichoke genomes after the WGT-1 event. However, sunflower and *M. micrantha* experienced another WGD event after WGT-1. First, the *Sc. taccada* genome was used as the common reference to generate orthologous regions relative to other selected Asteraceae genomes by MCscan[87]. For lettuce and artichoke, the three best matches to each *Sc. taccada* region were extracted to compute the layout for the gene-level equivalents. The genomic regions that contained consecutive triplicated homologous genes were defined as TRRs. For sunflower and *M. micrantha*, the six best matches to each *Sc. taccada* region were extracted, and the genomic regions that contained four, five, and six consecutive copies of homologous genes were defined as TRRs.

The genomic composition of the TRRs in the Asteraceae species were further investigated. The repeat sequences were integrated and annotated with Extensive de novo TE Annotator (EDTA, v2.0.0)[89]. The genomic composition (genic or repetitive sequences) within 1-Mb windows and 1 Mb steps across the genome was calculated for comparison. Pairwise data arrays were subjected to a two-tailed Student's *t*-test to examine the significance of difference. Each repetitive sequence family was subjected to a hypergeometric test to predict the enrichment or depletion in TRRs compared with their genome-wide distribution. A multiple testing correction was conducted by the false-discovery rate (FDR) method, and the adjusted threshold was set to $P < 0.01$.

Gene ontology (GO) was conducted in the TRRs using Cytoscape (v3.8.2)[90]. Significantly enriched GO terms ($P < 0.01$) shared by the genomes were identified and visualized using the ggplot2 package. The overrepresented gene families in the TRRs were examined using a hypergeometric test.

## Protein functional domain enrichment analysis

The function of the proteins encoded by all predicted genes was annotated using InterProScan (v5.24)[91] by searching publicly available database. The Gene Ontology (GO) and Kyoto Encyclopedia of Genes and Genomes (KEGG) IDs for each gene were assigned according to the corresponding InterPro entry. Enrichment analysis was performed based on the functional domains of all encoded proteins across the 29 species using the Fisher test with an FDR correction.

## Gene expression analysis

After clipping the adapter sequences and removing low-quality reads, the RNA-seq data from each sample were mapped to the reference genomes using Hisat 2 (v2.1.2)[59] and StringTie (v1.3.4)[92] with default parameters. Gene expression levels were normalized as fragments per kilobase of exon per million fragments mapped (FPKM).

## Plant transformation

The plant tissues used for the PII function study were from the lettuce cultivar 'Grand Rapids'. Surface-sterilized seeds were germinated on Murashige and Skoog (MS) agar plates containing 3% (w/v) sucrose and 0.8% (w/v) agar, with pH adjusted to 5.8 with KOH. The plants were grown in growth chambers at 25 °C under a 16-h light/8-h dark photoperiod. The full-length *PII* cDNA from carrot (*DcPII*, DCAR_002917), tomato (*SlPII*, Solyc06g009400.3.1) and Arabidopsis (*AtPII*, AT4G01900) was cloned into a plant expression vector (pYBA1132) respectively, and then transformed into lettuce with Agrobacterium

(*Agrobacterium tumefaciens*, strain EHA105) using the leaf disc transformation method[93].

## Reverse transcription quantitative PCR (RT-qPCR) analysis

Total RNA was extracted from lettuce seedlings using the TRIzol reagent (Invitrogen, Carlsbad, CA, USA) and treated with DNase I (TaKaRa, Dalian, China) to eliminate genomic DNA contamination. Total RNA was reverse transcribed into cDNA using SuperScript II Reverse Transcriptase (Invitrogen) and random primers (Promega, Madison, WI, USA). Relative *PII* expression was measured using Power SYBR Green PCR Master Mix (Applied Biosystems, Waltham, MA, USA) on an ABI 7500 thermocycler (Applied Biosystems) following the manufacturer's instructions. The primers were designed using Primer Premier 5 (http://www.premierbiosoft.com/primerdesign/) (Supplementary Data 15). Three biological replicates were performed for each sample. Lettuce *TUBULIN* (*TUB*) was used as an internal reference to normalize the data. The primers were *TUB*-F: 5′-TAGGCGTGTGAGT-GAGCAGT-3′ and *TUB*-R: 5′- AACCCTCGTACTCTGCCTCTT-3′. The fold-change in gene expression values was calculated using the $2^{-\Delta\Delta Ct}$ (cycle threshold) method. Relative gene expression values were plotted using SigmaPlot version 10.0 (SYSTAT Software, Inc., https://systatsoftware.com/).

## Subcellular localization analysis

The leaves of transgenic plants carrying the pYBA1132-derived construct expressing the green fluorescent protein (GFP) fusion were cut into small squares for fluorescence observation. The fluorescence from GFP or chloroplast autofluorescence was observed by confocal laser-scanning microscopy (ZEISS710; Carl Zeiss, Oberkochen, Germany).

## GST pull-down assay

The coding regions of *PII*s in three species and *LsNAGK* were ligated to pMal-C5x and pGEX-4T-1 vector, respectively. Constructed plasmids were transformed into *Escherichia coli* BL21 competent cells (ZOMANBIO, Beijing, China). Empty vectors also transformed to the competent cells for negative control. Cells were grown in Luria-Bertani (LB) medium at 37 °C until OD600 reached 0.5 then cooled to 16 °C. Add IPTG to the medium to the final concentration to 200 µM and cultured overnight in an incubator at 160 rpm at 16 °C. The MBP and MBP-fused PIIs protein was purified by Amylose Resin (NEB, MA, US). The GST and GST-fused LsNAGK protein was purified by GST Mag-Beads (Sangon Biotech, Shanghai, China).

Purified GST and GST-fused LsNAGK protein (10 nmol) were adsorbed onto the GST magnetic beads. MBP and MBP-fused PIIs proteins were added into the system and incubated at room temperature for two hours. Then the magnetic beads were washed twice and boiled in 1× SDS loading buffer for 15 min and analyzed by Western blot using anti-GST and anti-MBP antibody (Yeasen, Shanghai, China).

## Bimolecular fluorescence complementation

For bimolecular fluorescence complementation (BiFC) experiment, the constructed plasmids and empty vectors were transformed to the Agrobacterium EHA105 competent cells (Coolaber, Beijing, China). At the same time, Agrobacterium GV3101 carrying the P19 expression protein was cultured. Different combinations of cYFP, nYFP, and P19 agrobacterium solutions were co-infiltrated into the leaves of *Nicotiana benthamiana*. The fluorescence signals were detected by using Nikon A1 confocal microscope. The YFP signal is excited using a laser at 488 nm. We also detected chloroplast autofluorescence at 665 nm to determine the location of LsNAGK and PIIs.

## Metabolite detection

Fresh plants were harvested, immediately frozen, and ground into a fine power in liquid nitrogen. To determine the contents for free amino

acid monomers, samples were extracted with 0.1 M hydrochloric acid and derivatized with Waters AccQ•Tag reagent. The sample extracts were analyzed using an UPLC-Orbitrap-MS system (UPLC, Vanquish ultra-high performance liquid chromatography system; MS, Q Exactive hybrid Q−Orbitrap mass spectrometer, Thermo Fisher Scientific, USA). The total free amino acid contents were calculated as the sum of the contents of 25 free amino acid monomers.

ACCase activity was detected via the molybdenum blue method (Boxbio, Beijing, China). In brief, ACC can catalyze acetyl coenzyme A, NaHCO3 and ATP to generate malonyl CoA, ADP and inorganic phosphorus. Molybdenum blue and phosphate generate a substance with a characteristic absorption peak at 660 nm. ACC activity is determined by measuring the increase of inorganic phosphorus by ammonium molybdate phosphorus determination method. We measured the ACCase activity of wild-type and overexpressing *PII*s lettuce using BioTek Synergy H1 Multimode Microplate Reader (Agilent, Santa Clara, CA, USA) for three biological replicates and three technical replicates.

The nitrate nitrogen content was measured by the nitrosalicylic acid method using the corresponding assay kit (Boxbio, Beijing, China). In brief, $NO_3^-$ can react with salicylic acid to form nitrosalicylic acid under the condition of concentrated acid, which shows yellow under the condition of pH>12. Within a certain range, the color depth is proportional to the content. We measured the nitrate nitrogen content of wild-type and overexpressing *PII*s lettuce using BioTek Synergy H1 Multimode Microplate Reader (Agilent, Santa Clara, CA, USA) for three biological replicates and three technical replicates.

### Reporting summary
Further information on research design is available in the Nature Portfolio Reporting Summary linked to this article.

## Data availability
The sequencing data used in this study, assembled chromosomes, unplaced scaffolds, and annotations have been deposited into the Genome Sequence Archive (GSA) and Genome Warehouse (GWH) database in the BIG Data Center under accession code PRJCA007442. Annotated information on stem lettuce in detail can also be found in LettuceGDB [https://lettucegdb.com/][94]. Additional files including the customized repeat library, gene trees and phylogenetic trees have been uploaded to Zenodo [https://zenodo.org/record/8058114][95]. Source data are provided with this paper.

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

## Acknowledgements

We thank Professor Hongzhi Kong and Professor Guangyuan Rao for discussion and comments. Funding was provided by Beijing Academy of Agriculture and Forestry Sciences (KJCX201907-2, JKZX202201, and YXQN202203), Beijing Municipal Bureau of Agriculture and Rural Affairs (G20220628003-13), and the National Natural Science Foundation of China (31621001).

## Author contributions

X.Y., L.L., and J.W. conceived this study. Y.W. grew the plants and collected samples. F.S. performed the genome assembly, annotation, and data analyses. Y.Q., R.W., and X.H. generated and analyzed the lettuce PII transgenic lines. Y.J., Y.Z., J.H., and T.G. provided conceptual advice. X.Y., F.S., and L.L. wrote the manuscript with contributions from all the authors.

## Competing interests

The authors declare no competing interests.
