## [Peer Review File · Nature Communications]

Comparative Genomics Reveals a Unique Nitrogen-Carbon Balance System in AsteraceaeReviewers' Comments:

Reviewer #1:

Remarks to the Author:

- Suitable Quality?: Only in the Genomics part
- General Interest?: Yes
- Clearly Written?: Yes
- The noteworthy results?: Only in the Genomics analysis section
- Significance to the field and related fields?: Yes
- How does it compare to the established literature?: No; especially in the PII signaling
- Conclusions and claims Justified: No
- Additional evidence needed?: Yes
- Are there any flaws in the data analysis, interpretation and conclusions?: Yes; in the PII section
- Do these prohibit publication or require revision?: Yes
- Is the methodology sound?: Only in the Genomic section
- Does the work meet the expected standards in your field?: Yes; especially the work related to PII section
- Is there enough detail provided in the methods for the work to be reproduced?: More details are needed

A comprehensive piece of work, especially in the genomics and transcriptomics analysis section, that fills in a crucial part of Asteraceae evolution and propose for first time that the absence of the evolutionary loss of the PII signaling machinery in Asteraceae is the main reason for their ecological success to inhabit almost all ecological niches on Earth, making them among the top three invasive species worldwide. They propose a mechanism by evolving stepwise upgrades of the carbon/nitrogen balance system(s) via duplications events and enrichment of copy numbers of key C/N metabolic genes (e.g. nitrate transporters [NRT2/3], ADSs and FADs), resulting in enhanced fatty acid biosynthesis and nitrogen uptake. This work may inspire future studies to investigate the metabolic adaptation machinery of Asteraceae on physiological and biochemical levels.

Generally, the manuscript by Shen et al. reports an interesting series of data on the genomics and transcriptomics levels. They assembled the genome of lettuce (*Lactuca sativa*; a member of the Asteraceae) and of beach cabbage (*Scaevola taccada*; a member of the Goodeniaceae, which is phylogenetically the closest outgroup to Asteraceae). The authors compared and build on an already extensive sets of 7 sequenced Asteraceae (including lettuce) and 21 representatives of different phylogenetic branches of land plants, including coffee (*Coffea arabica*) and carrot (*Daucus carota*; another close outgroup to Asteraceae). In summary, the analysis revealed surprising results; an evolutionary loss of the PII signaling protein in Asteraceae and Goodeniaceae. In addition, they used the transcriptomic data from the 1KP Transcriptomic Initiative, including 39 Asteraceae species and Goodeniaceae species, which further supported the absence of PII from both families. Despite the solid genomic and transcriptomics analysis approaches and findings, the study still needs much work to establish more soundly the physiological implications the authors infer from their sequencing analysis, especially on the loss of PII machinery. Nevertheless, the manuscript is especially strong in providing the necessary genomics traits based on using sequencing technology for understanding the basis for the evolution of Asteraceae and the diversification compared to the other plants, leading to the ecological widespread of Asteraceae, yet the physiological roles of the PII signaling protein are wrongly interpreted. Despite attempts to verbally and experimentally describe the possible regulatory network of PII machinery, this crucial aspect of the work remains undeveloped and consequently the manuscript lacks the functional relevance. Scientifically speaking, I have some critical concerns on the physiological section of PII and I wish if they could be addressed first, as following:

Major Comments:

- The authors mentioned in several places the ref. #9, however it seems for me that they overlooked or misunderstood the take-home message of this key paper in PII research in plant-Kingdom. This paper shows that in course of evolution of plant PII proteins, they required a C-terminal extension (called Q-loop), which allowed the PII to sense and integrate directly the nitrogen availability in the cell, via binding glutamine. Glutamine binding to PII promotes the interaction and activation of N-acetyl-L-glutamate kinase (NAGK). The Q-loop motif is highly conserved in plants except in Brassicaceae family, to which *Arabidopsis thaliana* belongs. The Q loop in members of the Brassicaceae family exhibits a deletion, which made the *Arabidopsis* PII insensitive to glutamine. This finding was further confirmed by series of studies e.g. in *Polytomella parva* (Selim et al., 2020; FEBS J), *Chlorella variabilis* (Minaeva et al., 2015) and *Myrmecia incise* (Li et al., 2017). All of those informations were recently reviewed in Ref. #14 (<https://nph.onlinelibrary.wiley.com/doi/10.1111/nph.16492>)
- Therefore, an obvious question: What are the rational bases for choosing *Arabidopsis* PII protein to express it transgenically into lettuce? Why the authors did not choose a plant PII representative with unmutated Q-loop or from carrot as a close outgroup species of lettuce?
- PII acts downstream of GS-GOGAT cycle by regulating NAGK and thereby the arginine production, therefore it's not clear to me why glutamine and glutamate levels are affected negatively even if they are upstream of the PII effect. On another hand, PII stimulates positively NAGK activity, therefore arginine, citrulline and ornithine should be affected positively, if expressing foreigner "Arabidopsis PII" would stimulate NAGK activity.
- Only in one case, PII was able to inhibit NAGK activity in absence of glutamine and activate it in presence of glutamine in case of *Polytomella parva* (Selim et al., 2020; FEBS J), however here PII has the Q-loop extension, making it able to integrate the Gln levels of the cell. But again here, the authors chose the wrong PII, which does not sense glutamine, and thus the reduction of Gln, Glu, Arg does not make any sense.
- Finally, the reduction of the fatty acids levels makes sense in case of expressing a functional PII, which is able to interact and inhibit the ACCase, but the reduction of both the amino acids and the fatty acids levels seems to me a consequence of metabolic stress of expressing foreign protein.
- Targeting the chloroplast is not surprising as I assume that they used full length PII from *Arabidopsis*, including the chloroplast targeting sequence But, is it functional PII? how was it expressed under native or strong promoter? was the PII expressed as GFP fusion or as native form? Is the metabolomics analysis done using the PII-GFP fusion or not? all of those questions are not clear in the material and the method section, and could answer a lot of my previous questions, especially of the metabolic stress for expressing foreign PII protein.
- A key experiment, which I strongly suggest, is in vitro NAGK assay using lettuce-NAGK alone and on combination with *Arabidopsis*-PII and of carrot-PII, as described in Ref. #14. Also, I would like to see a multiple sequence alignment for *Arabidopsis*-PII and of carrot-PII and other members of canonical PII proteins, which contain canonical Q-loop motif (e.g. *Physcomitrella*, *Oryza*, *Chlamydomonas reinhardtii* and *Polytomella parva*).
- Figure 6 (A) encapsulates the essential problem with this manuscript: it discusses the specific mechanics of the PII protein and its interactions to form complexes with the transporter (NRT2/3; highly speculative as it's not proven for plants), ACCase, and NAGK, where Arg inhibits NAGK but not PII. PII is able to relieve NAGK from the Arg-feedback inhibition with complexing with NAGK.

Minor points:

- L107: "PII, which occurs widely in plants, animals, bacteria, and other organisms". This is wrong. Canonical PII proteins are not found in animals, only the PII-like proteins like CutA found in animals. What does the "other organisms" refer to?
- L108: those references 9-12 are so old, especially refs. 11-12. Reference 10 is not suitable here, can be replaced by ref. #14 or the original research "Feria Bourrellier AB, Valot B, Guillot A, Ambard-Bretteville F, Vidal J, Hodges M. Chloroplast acetyl-CoA carboxylase activity is 2-oxoglutarate-regulated by interaction of PII with the biotin carboxyl carrier subunit. Proc Natl Acad Sci U S A. 2010 Jan 5;107(1):502-7."

- L128-129: "Moreover, PII may participate in the negative regulation of N uptake in land plants¹⁴". This is not fully true as the nitrite uptake and sensitivity were shown to be higher in Arabidopsis PII knockout mutants than in the wild-type (Ferrario-Méry et al., 2005; Ferrario-Méry et al., 2008), implying that the PII-mediated regulation of nitrite uptake by Arabidopsis chloroplasts is similar to that by cyanobacteria (Watzer et al., 2019). This indicates that PII is needed to prevent overexcess of nitrite uptake.
- L131: "Consistent with previous reports^{9,11}". Completely wrong references here, refs. #9 & #10 clearly do not have anything about PII localization into chloroplast.
- Can you quantify the levels of the citrulline and ornithine?
- L136: "These results indicate that PII is absent in wild-type lettuce", Why? – this sentence is wired and misleading.
- L138: "this protein may be under selective pressure during evolution⁹" >> wrong reference.
- L156-158: Why the duplication event is then needed?. If PII is present and reduce N-uptake then it's loss would give directly an uptake advantage and then they would not need duplication events of the transporters. Another thought that the authors did not consider that PII is needed to control the N-flow into the cell to avoid extra toxicity of nitrite and ammonia accumulation.
- L160-164: Can be validated using RT-PCR for fatty acid genes?
- L173-174: "PII is subject to negative feedback to maintain N homeostasis" ... This is not clear to me, what do the authors mean?
- L174-176: losing PII from the Goodeniaceae would increase the fatty acid biosynthesis by relieving the ACCase from PII inhibition.
- L185-187: This sounds as removal of PII would be beneficial to the cells, if the authors mean that this clearly a wrong statement because in absence of PII the entire metabolism would be missed up.
- L270: are those metabolites quantified in different transgenic lines? or single line? and was PII GFP-fused or native?
- L437-444: the RNA-seq data was used for what? this is not clear through the manuscript.
- L532-535: not clear if PII was fused to GFP which was used for the localization study and then used for metabolomics analysis or PII without GFP was used for the metabolomics analysis.
- L540: "relative PII expression was measured ...", which Figure? not clear
- L556-565: if the authors used a PII-GFP fusion as I assume, how did they confirm that the PII is functional? – there is a lot of indications that the GFP could hinder the full function of PII, especially for the interaction with ACCase, due to the hindrance.
- L563: I am not familiar with this method to measure the ACCase activity, can you describe it briefly?

Supplementary note:

- L1115-1116: I am not aware the PII influence on Glu and Gln biosynthesis, which ref. do you mean? and what do you mean by negative feedback of arginine? "Not clear"
- L1111-1121: Can the authors confirm by any biophysical or biochemical methods that the lettuce NAGK is still interacting with AraPII? using pulldown or size exclusion chromatography or surface plasmon resonance or any relative method.

Reviewer #2:

Remarks to the Author:

Review of Comparative Genomics Reveals a Unique Nitrogen-Carbon Balance System in Asteraceae

This manuscript presents the results of two newly sequenced genomes and provides a comparative genomics analysis with other Asteraceae genomes. They posit that a whole genome duplication event in early in the history of Asteraceae evolution set the stage for the evolution of a novel nitrogen-carbon balance system in the family. They demonstrate that PII, which has a significant role in sensing and regulating nitrogen-carbon signals was lost in the ancestor of Asteraceae and Goodeniaceae. They hypothesize that whole genome duplication events in Asteraceae led to the

expansion of high-affinity nitrate transporter genes and fatty acid biosynthesis genes thereby allowing the family to be more evolutionarily successful. The paper was thorough in general its methods and analyses with some points noted below. The work is very thorough, and the addition of functional analyses is noteworthy. The major concern I had with the thesis is that the authors failed to discuss any other reasons in the main body of the manuscript (though it was touched on in the notes) why Asteraceae has been so successful and the current presentation of the work seems to hinge on this one aspect. It is striking and impactful that there are no PII genes and the paper offers a very cool demonstration of how the gene could have been lost. And the expansion of specific classes of genes is also compelling, but still the argument correlative and overreaches in its current state. Is there evidence that such a nitrogen-carbon balance system would be a priori hypothesized to lead to ecological and evolutionary success. This was not addressed in the otherwise very thorough notes either. If so, the strength of this story would be greatly enhanced. What about lineages within Asteraceae that are not successful, for example, the Barnadesioideae. Additionally, Barker et al. show a duplication even shared with Calyceraceae and Asteraceae, plus the one that is unique to Asteraceae. This lack of sampling here means there is a missing link with Calyceraceae. While it could still be the Asteraceae-specific WGT, there should be more discussion of the lack of sampling and what that means for the study conclusions. Please also note that I am not an expert in the nitrogen-carbon balance system, so while I found their functional work compelling, I can't speak in depth to the details and the rationale behind their conclusions drawn.

Additional Comments:

Lines 41-44: This statement is not true as per their phylogeny in Figure 1. "Here, we generated high-quality genome assemblies for stem lettuce (*Lactuca sativa* var. *angustana*), a member of the Asteraceae, and beach cabbage (*Scaevola taccada*), a representative of the Goodeniaceae family, which is phylogenetically the closest outgroup to the Asteraceae (Fig. 1a,b)."

Lines 62-73: Are there some typos in this paragraph. WGT-2 is not cited but seems to be discussed? As written, the description of triplication events is unclear.

Line 430, where is the customized repeat library available? GitHub?

Line 462, was model selection used to choose this model?

Line 467, where are the gene trees or matrices used to generate these phylogenies?

Line 476, where are the data on the fossils that were used?

Line 535, what are the tissue culture/regeneration methods/details?

Line 563, cite manufacturer details.

Line 569, the link does not work. Please ensure all these data that are stated are deposited.

Figure 2a, readability would be improved by ordering the nodes as is more traditional in phylogenies.

Figure 2b, consider reorganizing taxa list perhaps phylogenetically.

Figure 4e, the colors corresponding to the taxa look different from the legend, e.g, legend *H. annuus* is darker than in the figure.

There are 39 supplemental tables but they are not mentioned in the body of the manuscript.

Data for the selection analyses on the genes is absent from the manuscript and the supplementals,

please include.

Reviewer #3:

Remarks to the Author:

In the study entitled "Comparative Genomics Reveals a Unique Nitrogen-Carbon Balance System in Asteraceae", Shen and colleagues constructed two chromosome-scale genomes for lettuce (*Lactuca sativa* var. *angustana*, a member of the Asteraceae) and *Scaevola taccada*, (a member of 22 the closest outgroup, the Goodeniaceae). They further performed comparative genomics analysis for 29 representative terrestrial plant species, and deduced that Asteraceae was originated from the paleopolyploidization event which occurred ~80 MYA. Notably, the detail comparative genomics analysis revealed that the Asteraceae genomes absence PII, the universal regulator of nitrogen-carbon (N25 C) assimilation present in almost all domains of life. They thus proposed that the Asteraceae evolved a unique N-C balance system following the loss of PII, resulting in enhanced N uptake capacity and fatty acid biosynthesis. This study has ground-breaking for the evolution of Asteraceae. The manuscript was well organized. I only have few comments:

1. The basic information of assembled genome should be presented in the main text, for example: "Line 46-48: "The genome assembly of *Sc. taccada* (ST1.0) contained 8 pseudo-chromosomes and covered 1,159 Mb with detailed annotations (Extended Data Fig. 2; Supplementary Note 3)." The annotated gene number and the length of N50 should be presented.
2. Fig. 3b, please label the centromere position on chromosomes.
3. Extended Data Fig. 2, No. of Sequences should be No. of contigs. HiC should be Hi-C. Effective number should be kept consistent, for example: Complete BUSCO (%) 95.42 (*La.sativa*) and 95.2 (*Sc.taccada*)

The author may need to reformat for Nature communications. I would suggest the editor to accept the manuscript after mirror revision.

REVIEWER COMMENTS

Reviewer #1 (Remarks to the Author):

- Suitable Quality?: Only in the Genomics part
- General Interest?: Yes
- Clearly Written?: Yes
- The noteworthy results?: Only in the Genomics analysis section
- Significance to the field and related fields?: Yes
- How does it compare to the established literature?: No; especially in the PII signaling
- Conclusions and claims Justified: No
- Additional evidence needed?: Yes
- Are there any flaws in the data analysis, interpretation and conclusions?: Yes; in the PII section
- Do these prohibit publication or require revision?: Yes
- Is the methodology sound?: Only in the Genomic section
- Does the work meet the expected standards in your field?: Yes; especially the work related to PII section
- Is there enough detail provided in the methods for the work to be reproduced?: More details are needed

A comprehensive piece of work, especially in the genomics and transcriptomics analysis section, that fills in a crucial part of Asteraceae evolution and propose for first time that the absence of the evolutionary loss of the PII signaling machinery in Asteraceae is the main reason for their ecological success to inhabit almost all ecological niches on Earth, making them among the top three invasive species worldwide. They propose a mechanism by evolving stepwise upgrades of the carbon/nitrogen balance system(s) via duplications events and enrichment of copy numbers of key C/N metabolic genes (e.g. nitrate transporters [NRT2/3], ADSs and FADs), resulting in enhanced fatty acid biosynthesis and nitrogen uptake. This work may inspire future studies to investigate the metabolic adaptation machinery of Asteraceae on physiological and biochemical levels.

Response: we really appreciate all the comments the Reviewer made, and thanks a lot for recognizing our discovery. We also believe that this work will lead to more research that

understands the rich diversity and excellent environmental adaptability of Asteraceae plants from physiological and biochemical perspectives.

Generally, the manuscript by Shen et al. reports an interesting series of data on the genomics and transcriptomics levels. They assembled the genome of lettuce (*Lactuca sativa*; a member of the Asteraceae) and of beach cabbage (*Scaevola taccada*; a member of the Goodeniaceae, which is phylogenetically the closest outgroup to Asteraceae). The authors compared and build on an already extensive sets of 7 sequenced Asteraceae (including lettuce) and 21 representatives of different phylogenetic branches of land plants, including coffee (*Coffea arabica*) and carrot (*Daucus carota*; another close outgroup to Asteraceae). In summary, the analysis revealed surprising results; an evolutionary loss of the PII signaling protein in Asteraceae and Goodeniaceae. In addition, they used the transcriptomic data from the 1KP Transcriptomic Initiative, including 39 Asteraceae species and Goodeniaceae species, which further supported the absence of PII from both families. Despite the solid genomic and transcriptomics analysis approaches and findings, the study still needs much work to establish more soundly the physiological implications the authors infer from their sequencing analysis, especially on the loss of PII machinery. Nevertheless, the manuscript is especially strong in providing the necessary genomics traits based on using sequencing technology for understanding the basis for the evolution of Asteraceae and the diversification compared to the other plants, leading to the ecological widespread of Asteraceae, yet the physiological roles of the PII signaling protein are wrongly interpreted. Despite attempts to verbally and experimentally describe the possible regulatory network of PII machinery, this crucial aspect of the work remains undeveloped and consequently the manuscript lacks the functional relevance. Scientifically speaking, I have some critical concerns on the physiological section of PII and I wish if they could be addressed first, as following:

Response: Again, thank the Reviewer very much for the professional comments and valuable suggestions, in particular the appreciation on our findings in omics parts. We are also very happy that you put forward many valuable suggestions to expand our understanding of the function of PII and its evolution in plants.

As the Reviewer pointed out and suggested, we carefully scanned all PII-related studies in plants and tried our best to provide more evidence and meanwhile adjusted our conclusions partially. For instance, we added the transgenic studies with carrot PII (*Dc-PII*) and tomato PII (*Sl-PII*), both with complete Q-loop, into lettuce and followed physiological tests. We also carried out a series of *in vivo/in vitro* experiments to validate interactions between PIIs (from Arabidopsis, carrot, and tomato, respectively) and NAGK (from lettuce). Meanwhile, we adjusted our model such as correcting the negative feedback of Arg to NAGK instead of PII. Taken together, we hope the Reviewer will find these revisions satisfactory. A point-to-point response is as follows.

Major Comments:

- The authors mentioned in several places the ref. #9, however it seems for me that they overlooked or misunderstood the take-home message of this key paper in PII research in plant-Kingdom. This paper shows that in course of evolution of plant PII proteins, they required a C-terminal extension (called Q-loop), which allowed the PII to sense and integrate directly the nitrogen availability in the cell, via binding glutamine. Glutamine binding to PII promotes the interaction and activation of N-acetyl-l-glutamate kinase (NAGK). The Q-loop motif is highly conserved in plants except in Brassicaceae family, to which *Arabidopsis thaliana* belongs. The Q loop in members of the Brassicaceae family exhibits a deletion, which made the *Arabidopsis* PII insensitive to glutamine. This finding was further confirmed by series of studies e.g. in *Polytomella parva* (Selim et al., 2020; FEBS J), *Chlorella variabilis* (Minaeva et al., 2015) and *Myrmecia incise* (Li et al., 2017). All of those informations were recently reviewed in Ref. #14 (<https://nph.online-library.wiley.com/doi/10.1111/nph.16492>). Therefore, an obvious question: What are the rational bases for choosing *Arabidopsis* PII protein to express it transgenically into lettuce? Why the authors did not choose a plant PII representative with unmutated Q-loop or from carrot as a close outgroup species of lettuce?

Response: We greatly appreciate your professionalism and in-depth knowledge of the field of PII research. We have benefited from the extensive literature and explanation of the PII features that you have provided. In our previous version, in order to confirm the role of PII in carbon and nitrogen balance, and to speculate about the potential impact of PII loss in Asteraceae species, we performed the transformation of PII genes in lettuce. The reason why we chose the *Arabidopsis PII (At-PII)* gene as the foreign one for genetic verification is to consider that as the model of higher plants, the functional studies of *At-PII* are the most abundant (Hsieh *et al.*, 1998; Smith *et al.*, 2003; Ferrario-Méry *et al.*, 2005, 2006; Mizuno *et al.*, 2007b,a; Baud *et al.*, 2010; Bourrellier *et al.*, 2010). For example, there are overexpression and knock-out (mutant) studies of *At-PII*, respectively (Hsieh *et al.*, 1998; Ferrario-Méry *et al.*, 2005, 2006; Baud *et al.*, 2010; Bourrellier *et al.*, 2010). Therefore, we thought our transgenic studies can be compared with these studies side-by-side such as the wild-type lettuce *versus* the PII mutant of *Arabidopsis* and the overexpression of *At-PII* lettuce *versus* the wild-type or the overexpression of *At-PII* *Arabidopsis*. Taken the phenotypic analysis of PII overexpressors in both lettuce and *Arabidopsis* as an example, as Hsieh *et al.*, 1998 (PNAS) did, we also tried to observe whether the anthocyanin content was changed under different N resources. Unfortunately, we did not observe the same phenotypic changes as in *Arabidopsis*. We attached more discussion in Supplementary Note 7.2 in “In-depth discussion and interpretation” to explain the side-by-side comparison.

Besides, we totally agree with the Reviewer that because of the PII of *Arabidopsis* with a mutant Q-loop leading to its insensitivity to glutamine, the *At-PII* is indeed not a perfect choice. However, as you can see in our paper, the overexpression of the *Arabidopsis* PII gene in lettuce does have a substantial physiological impact, which once again verifies the importance of PII in carbon and nitrogen balance, and also shows that Asteraceae represented by lettuce have potentially developed a unique carbon and nitrogen balance system in the absence of PII.

As you suggested, we added experiments of transforming Dc-PII and Sl-PII genes into lettuce, respectively. First of all, sequence alignments confirmed that PII genes in carrot and tomato have complete Q-loops (Fig. S49; Supplementary Notes 7.2). Then, after obtaining the

transgenic plants, a series of physiological tests were performed, and we presented them with each question raised by the Reviewer as follows.

Polytomella parva	--MNALNFSPTLFNKQT----ISPTRQT-AFITVQGTVR--KAK--SNFSTSIRQP--LV	47
Chlamydomonas reinhardtii	MALASRTSSAAVVGRI-----STRSA-AVVPVRSIAS--RCQ--A--ARPARRASVAV	45
Physcomitrella patens	MALQPRLSL.SCLRGRSVDACAFAV-PASASISASSDACVRIPCWNGASSSS--KRLPFFG-	57
Oryza sativa	MS-----SPATAA-----AAAACSGVLRHHHPAS--PRPPPTTTT	34
Arabidopsis thaliana	MA-----ASMTKPI.SITSLGFYSDRKNIAFSDCISICSGFRH-----SRPS---	41
Solanum lycopersicum	MA-----SPSLSKSNFSLHSFSSSPSLSQFPHFTSITVVQPK-----FFPS---	41
Daucus carota	-----SPSLSKSNFSLHSFSSSPSLSQFPHFTSITVVQPK-----FFPS---	34
Polytomella parva	-----VV-SAAKSDAPFK--RATYDDLES IKANLSAFPSCFEFFRVE	85
Chlamydomonas reinhardtii	-----RA-SDENGVSVR--RATYAELESIQCDSLAFPGVKFFRIE	83
Physcomitrella patens	-----ARVASADP--KSPNWRKRVSGVVQVHLEEDFDQSKDYQPSVDFYKVE	105
Oryza sativa	TTSRLLASRSRGLQRPLRVNHAPRRLPP-----TAARAQSAAGYQPESEFYKVE	87
Arabidopsis thaliana	-CLDL-----VTKSPSNNSRV-----LPVVSQAQISS-DYIPDSKFYKVE	78
Solanum lycopersicum	---Q-----LTFKRCQNAPS-----FPIIRAQNSP-DFVPDAKFYKVE	75
Daucus carota	TSSSR-----FSLKFSKISPL-----SPVIRAQSAPTEDFPDAKFYKVE	73
Polytomella parva	AVIRPWRLPFVVEQLGNNIRGMTVTSVHGIQGGSRERYGGTEFSQTDLVEKQKVEIV	145
Chlamydomonas reinhardtii	AIFRPWRLPFVIDTL.SKYGIRGLTNPVKGVGQGGSRERYAGTEFGPSNLVDKEKLDIV	143
Physcomitrella patens	AVLRPWRLSPVSSALLKMGIRGVTVDVVRGFGAQQGSRERQAGTEYAGDSYLKVKLEIV	165
Oryza sativa	AILRPWRVPPYSSGLLQMGIRGVTVDVVRGFGAQQGSTERHEGSEFAEDTFIDKVKMEIV	147
Arabidopsis thaliana	AIVRPWRIQQVSSALLKIGIRGVTVDVVRGFGAQQGSTERHGGSEFSEDKFVAKVKMEIV	138
Solanum lycopersicum	AILRPWRVPPYSSALLKMGIRGVTVDVVRGFGAQQGLTERQAGSEFSEDTFVAKVKMEIV	135
Daucus carota	AIIRAWRPVKSLLALRMGIRGVTVDVVRGFGSQGGMKERHAGSEFGEEDMFVSKVKMEIV	133
Polytomella parva	VTRAQANIVRSRIATAAFTGEIGDGKIFIHPVAEVIIRIETGFLAEHMAGGEMDMMAS	205
Chlamydomonas reinhardtii	VSRAQVDVAVRLVAASAYTGEIGDGKIFVHPVAEVRIRIETGLEAEKMEGGMEDMMKK	203
Physcomitrella patens	VSKDQVEAVIDTIIIDQARTGEIGDGKIFVSPVSDIIRIIRIETGERGLKAERMAGGRAAMQTS	225
Oryza sativa	VSKDQVEAVVDKIIIEKARTGEIGDGKIFLIPVSDVIRIIRIETGERGERAERMAGGLADKLSS	207
Arabidopsis thaliana	VKKDQVESVINTIIEGARTGEIGDGKIFVLPVSDVIRVIRIETGERGEKAEMKMTG--DMLSP	196
Solanum lycopersicum	VSKDQVEGVIAKIIIEEARTGEIGDGKIFLTPISDVIRVIRIETGERGEKAERMGGHADMSSA	195
Daucus carota	VCKDQVEAVIEKIIIEEARTSQIGDGKIFVIPVADIIRVIRIETGERGEKAERMGGFRFDMSSS	193
Polytomella parva	KSTA---	209
Chlamydomonas reinhardtii	KK-----	205
Physcomitrella patens	AEGSDGN	232
Oryza sativa	AMPIS--	212
Arabidopsis thaliana	S-----	196
Solanum lycopersicum	LSTS---	199
Daucus carota	EA-----	195

Figure S49. Sequence alignment of PII of plants and green algae. Color labeling shows secondary structure assignment (strands, green; helices, yellow) according to the AtPII-AtNAGK complex structures. The truncated Q-loop part in Arabidopsis is highlighted by a red box.

- PII acts downstream of GS-GOGAT cycle by regulating NAGK and thereby the arginine production, therefore it's not clear to me why glutamine and glutamate levels are affected negatively even if they are upstream of the PII effect. On another hand, PII stimulates positively NAGK activity, therefore arginine, citrulline and ornithine should be affected positively, if expressing foreigner "Arabidopsis PII" would stimulate NAGK activity.

• Only in one case, PII was able to inhibit NAGK activity in absence of glutamine and activate it in presence of glutamine in case of *Polytomella parva* (Selim et al., 2020; FEBS J), however here PII has the Q-loop extension, making it able to integrate the Gln levels of the cell. But again here, the authors chose the wrong PII, which does not sense glutamine, and thus the reduction of Gln, Glu, Arg does not make any sense.

Response: Thank the Reviewer for raising these two key questions. Since they are closely related, we answered them together.

In fact, we agree with the Reviewer that expressing foreigner “PII” would stimulate NAGK activity and further increase the content of arginine, citrulline, and ornithine. However, our observation was not as we expected when we expressed *At-PII* in lettuce (The content of arginine, glutamine and glutamic acid was affected negatively (Figs. S51d-g).

As you suggested, we newly overexpressed *PIIs* from carrot and tomato into lettuce, respectively. We found that these foreigner *PIIs* could increase the synthesis of arginine (and glutamine, glutamic acid) in lettuce (Fig. S52c,d), which was different from that observation of expressing *At-PII*. The possible reason that caused the differences might be the distinct Q-loop regions of PII in carrot/tomato and Arabidopsis. Another possible reason is that we observed the nitrate contents of transgenic plants were all affected negatively, resulting in the reduction of N resources (Figs. S 51h, S52c,d).

In addition, we would like to point out that the loss event of PII occurred in tens of millions years ago, even before the origin of Asteraceae (~80 MYA), which potentially provided Asteraceae species enough time to evolve into a distinct C-N balance machinery. By now, we know very little about this machinery. Thus, some physiological and biochemical results in the transgenic lettuce plants maybe not be expected and be different from the plants with PII signaling machinery.

Figure S51. Functional study of expressing *AtPII* in lettuce.

a, Validation of chloroplast localization of PII in lettuce plants heterologously expressing Arabidopsis PII. We used confocal microscopy to visualize the localization of the PII-GFP (green fluorescent protein) fusion protein compared to that of free GFP in mesophyll protoplasts prepared from transgenic lettuce. Scale bars, 10 μ m. **b**, Bimolecular fluorescence complementation (BiFC) analysis showing the interaction between LsNAGK and At-PII *in vivo*. **c**, GST pull-down assays showing the interaction between LsNAGK and At-PII *in vitro*. The proteins were detected by Western blot analysis with anti-MBP and anti-GST antibodies. **d-g**, Quantification of total free amino acids (**d**), glutamic acid (**e**), arginine (**f**), glutamine (**g**), in wide-type and At-PII expressing lettuce plants. WT: wide-type; PII: different lines of PII-expressing lettuce. **h-i**, Quantification of nitrate nitrogen (**h**), and ACCase enzyme activity (**i**) in wild-type and PII-expressing lettuce. L1, L2, and L3 represent three independent transgenic lines. Statistical significance was determined using a Student's t-test (three independent assays). ** $P < 0.01$, *** $P < 0.001$, **** $P < 0.0001$.

a**c****d**
Figure S52. Functional study of expressing *DcPII* and *SIPII* in lettuce.

a, Bimolecular fluorescence complementation (BiFC) analysis showing the interaction between LsNAGK and PII (SIPII) *in vivo*. **b**, GST pull-down assays showing the interaction between LsNAGK and PII (SIPII) *in vitro*. The proteins were detected by Western blot analysis with anti-MBP and anti-GST antibodies. **c**, Quantification of total free amino acids, glutamic acid, arginine, glutamine, nitrogen, and ACCase enzyme activity in wide-type and *DcPII*-expressing lettuce plants. **d**, Quantification of total free amino acids, glutamic acid, arginine, glutamine, nitrogen, and ACCase enzyme activity in wild-type and *SIPII*-expressing lettuce plants. L1, L2, and L3 represent three independent transgenic lines. WT, wild-type; EV, empty vector control; Statistical significance was determined using a Student's t-test (three independent assays). ** $P < 0.01$, *** $P < 0.001$, **** $P < 0.0001$.

- Finally, the reduction of the fatty acids levels makes sense in case of expressing a functional PII, which is able to interact and inhibit the ACCase, but the reduction of both the amino acids and the fatty acids levels seems to me a consequence of metabolic stress of expressing foreign protein.

Response: Thank you very much for your question. We also tested ACCase in transgenic lettuce plants with *DcPII* and *SIPII* genes, and the results showed that ACCases were obviously inhibited (Figs. S51i, S52c,d), which was consistent with the result of *AtPII* transgenic plants.

- Targeting the chloroplast is not surprising as I assume that they used full length PII from Arabidopsis, including the chloroplast targeting sequence But, is it functional PII? how was it expressed under native or strong promoter? was the PII expressed as GFP fusion or as native form? Is the metabolomics analysis done using the PII-GFP fusion or not? all of those questions are not clear in the material and the method section, and could answer a lot of my previous questions, especially of the metabolic stress for expressing foreign PII protein.

Response: We apologize for the lack of supplemental materials we provided to the description in the previous version. In the updated version, we have made more detailed additions to the experimental methods and materials.

First of all, in *At-PII* and subsequent *Dc-PII*, *SI-PII* related genetic transformation experiments, we used full-length PII sequences. To demonstrate the function of these PIIs, we conducted a series of *in vitro* and *in vivo* tests (BiFC, Pull-down). Experiments have shown that PIIs of these three species can interact with NAGK in lettuce *in vitro* and *in vivo* (Figs. 3e, 3f, S51a-c, S52a,b). Therefore, we think the PIIs transformed into lettuce can play their biological function.

Secondly, the promoter we used in this study is the 35S promoter, because the PII gene is not present in lettuce and other Asteraceae species, so the original promoter in Asteraceae cannot be obtained. Direct use of the original promoter of Arabidopsis (or carrot/tomato) PII does not guarantee that the expression initiation/pattern of PII is correctly in lettuce. In order to ensure the expression of PII, we selected the 35S promoter that was used most frequently in the

genetic transformation experiments. Furthermore, in order to exclude or reduce the interference of you called "metabolic stress for expressing foreign PII protein" on the tests, we set up controls with empty vectors in subsequent experiments.

Thirdly, the transgenic plants with At-PII are two types. One type is with GFP fusion since we would like to see whether the foreign PII gene functions in Chloroplast. Another type are not with GFP, which are used to do the physiological and metabolic tests. The Dc-PII and Sl-PII transgenic plants are not fused with GFP, which are further employed to do physiological tests (including amino acid content, ACCase and nitrate content tests).

- A key experiment, which I strongly suggest, is *in vitro* NAGK assay using lettuce-NAGK alone and on combination with Arabidopsis-PII and of carrot-PII, as described in Ref. #14. Also, I would like to see a multiple sequence alignment for Arabidopsis-PII and of carrot-PII and other members of canonical PII proteins, which contain canonical Q-loop motif (e.g. *Physcomitrella*, *Oryza*, *Chlamydomonas reinhardtii* and *Polytomella parva*).

Response: As you suggested, we first performed multiple sequence alignments including PII genes from Arabidopsis, carrot, tomato, *Physcomitrella*, *Oryza*, *Chlamydomonas reinhardtii* and *Polytomella parva*, and the result clearly indicated that carrot PII and tomato PII have a complete Q-loop region (Fig. S49).

As we described previously, a series of *in vitro* and *in vivo* interaction tests, including BiFC and Pull-down, were performed. All the results supported that At-PII, Dc-PII, and Sl-PII could interact with lettuce-NAGK (Figs. S51b,c, 3e,f, S52a,b).

- Figure 6 (A) encapsulates the essential problem with this manuscript: it discusses the specific mechanics of the PII protein and its interactions to form complexes with the transporter (NRT2/3; highly speculative as it's not proven for plants), ACCase, and NAGK, where Arg inhibits NAGK but not PII. PII is able to relieve NAGK from the Arg-feedback inhibition with complexing with NAGK.

Response: Thank you very much for your suggestion. Whether there exists an interaction between PII proteins and the transporters (NRT2/3), we did not have direct evidence. Therefore, we used a dotted line in Figure 6 (A) to describe this biological process that has not been directly demonstrated. We are sorry that we did not described this in the figure legend, and explained with more details in the updated version.

Besides, we changed Arg-feedback inhibition to NAGK instead of PII in our model.

Minor points:

- L107: “PII, which occurs widely in plants, animals, bacteria, and other organisms”. This is wrong. Canonical PII proteins are not found in animals, only the PII-like proteins like CutA found in animals. What does the “other organisms” refer to?

Response: Revised accordingly. We change the sentence to “PII occurs widely in all three domains of life”.

- L108: those references 9-12 are so old, especially refs. 11-12. Reference 10 is not suitable here, can be replaced by ref. #14 or the original research “Feria Bourrellier AB, Valot B, Guillot A, Ambard-Bretteville F, Vidal J, Hodges M. Chloroplast acetyl-CoA carboxylase activity is 2-oxoglutarate-regulated by interaction of PII with the biotin carboxyl carrier subunit. Proc Natl Acad Sci U S A. 2010 Jan 5;107(1):502-7.”

Response: We have adopted the suggestions and revised them accordingly.

- L128-129: “Moreover, PII may participate in the negative regulation of N uptake in land plants¹⁴”. This is not fully true as the nitrite uptake and sensitivity were shown to be higher in Arabidopsis PII knockout mutants than in the wild-type (Ferrario-Méry et al., 2005; Ferrario-Méry et al., 2008), implying that the PII-mediated regulation of nitrite uptake by Arabidopsis chloroplasts is similar to that by cyanobacteria (Watzler et al., 2019). This indicates that PII is needed to prevent overexcess of nitrite uptake.

Response: Thank the Reviewer for pointing out this issue. We agree with the Reviewer that currently there is no direct evidence to support PII negatively regulating N uptake in land plants. As we mentioned previously, we observed the nitrate contents in our transgenic plants were down-regulated even though we have not known the exact reason.

- L131: “Consistent with previous reports^{9,11}”. Completely wrong references here, refs. #9 & #10 clearly do not have anything about PII localization into chloroplast.

Response: We have adopted the suggestions. The ref #9 is removed here and the right reference is added.

- Can you quantify the levels of the citrulline and ornithine?

Response: Unfortunately, our current assays can't be used to quantify the levels of citrulline and ornithine. Since these two amino acids are unstable, the determination method of stable amino acids can't be employed to measure these two amino acids accurately.

- L136: “These results indicate that PII is absent in wild-type lettuce”, Why? – this sentence is wired and misleading.

Response: Thanks a lot for this good catch. Based on the transgenic study, we would like to drive the point that “these results further confirmed the PII loss in lettuce”. We have reorganized the sentences in the revised manuscript.

- L138: “this protein may be under selective pressure during evolution⁹” >> wrong reference.

Response: This wrong reference has been removed.

- L156-158: Why the duplication event is then needed?. If PII is present and reduce N-uptake then it's loss would give directly an uptake advantage and then they would not need duplication events of the transporters. Another thought that the authors did not consider that PII is needed to control the N-flow into the cell to avoid extra toxicity of nitrite and ammonia accumulation.

Response: Thank the Reviewer for raising this question. Based on our current understanding, we don't know whether it is directly correlated between the whole genome triplication event (leading to the increase of *NRT2/3* genes) and PII-loss. In our study, a series of genomic analyses showed preferential retention and later tandem duplication of plant carbon and nitrogen-related genes (*NRT2/3*, *FADs*, *KASs*, *ADSs*), suggesting the need for a unique carbon and nitrogen balance during re-diploidization after polyploidization.

Meanwhile, as we pointed out previously, the loss event of PII occurred in tens of millions years ago, even before the origin of Asteraceae (~80 MYA). During the long history, Asteraceae species have potentially evolved into a distinct C-N balance machinery. By now, we know very little on this machinery, and our study is just a beginning. We hope our study can lead to more research to investigate the metabolic adaptation machinery of Asteraceae.

In addition, as another Reviewer pointed out that the potential reason of the ecological success of Asteraceae plants might not be only caused by the unique nitrogen and carbon balance system. WGT-1 event, dynamic repetitive sequences (especially LTRs), enriched neofunctionalization and subfunctionalization of gene families as we described in the revised manuscript and Supplementary Notes, are all underlying reasons.

- L160-164: Can be validated using RT-PCR for fatty acid genes?

Response: As you suggested, we selected several fatty acid genes randomly and confirmed the gene expression levels using RT-qPCR. All the results have been included in the supplementary Notes (Fig. S65).

Figure S65. Real-Time qRT-PCR of the duplicated *FAD* genes in lettuce.

- L173-174: “PII is subject to negative feedback to maintain N homeostasis” ... This is not clear to me, what do the authors mean?
- L174-176: losing PII from the Goodeniaceae would increase the fatty acid biosynthesis by relieving the ACCase from PII inhibition.
- L185-187: This sounds as removal of PII would be beneficial to the cells, if the authors mean that this clearly a wrong statement because in absence of PII the entire metabolism would be missed up.

Response: Thank the Reviewer for raising these questions/points. Since the above three points are on the C-N balance model in Asteraceae, we respond to them together.

In the steady state during genome evolution, PII regulates and maintains the carbon/nitrogen balance in plants by directly or indirectly regulating NAGK, ACCase, etc., senses the nitrogen level of plants and prevents excessive nitrogen absorption to cause toxicity (Hsieh *et al.*, 1998; Ferrario-Méry *et al.*, 2006; Baud *et al.*, 2010; Bourrellier *et al.*, 2010). In this balanced and stable genetic background, if we knock out, theoretically, there will be a series of expected results including: the loss of PII’s regulation of carbon/nitrogen balance, and the loss of PII’s inhibitory effect on plant excess nitrogen uptake, the loss of PII’s inhibitory effect on ACCase, so the entire metabolic balance will be broken and missed up. For plants with PII signaling machinery, knocking out this gene is completely negative and catastrophic to the metabolic stability and survival of plants.

In our study, we found that the loss event of this gene occurred in the Asteraceae ancestral genome, and it must be mentioned that this event was very old, even before the origin of Asteraceae (~80 Mya), which allowed plants to evolve new mechanisms, and in fact plants have to evolve mechanisms to adapt to the negatively impacted state of PII loss in order to survive under natural conditions. Therefore, the genetic background of Asteraceae plants is different from that of plants with PII signaling machinery. Although the results of genetic transformation experiments show that the overexpression of PII in lettuce can reconstruct the PII signaling machinery to a certain extent, this does not mean that Asteraceae plants have a genetic state that is more suitable for natural conditions.

An ancient polyploidization event occurred in Asteraceae plants at a time close to the PII loss event. The occurrence of ancient polyploidy events provided a genomic basis for the rapid evolution of plants. In our study, a series of genomic evidence showed that plant carbon metabolism, nitrogen metabolism, and related genes underwent preferential retention and tandem duplication (Supplementary Note 4.3), suggesting the need for a unique carbon and nitrogen balance during re-diploidization after polyploidy.

The in-depth excavation and comprehensive system analysis of plant carbon and nitrogen balance system is a systematic project. We found that PII signaling machinery, which is widely present in plants, was lost in Asteraceae plants. A relatively reasonable model has been proposed, but the improvement of this model is a long-term process. Similarly, the comprehensive understanding the rich diversity and super adaptability of Asteraceae plants is also need a long way to go.

- L270: are those metabolites quantified in different transgenic lines? or single line? and was PII GFP-fused or native?

Response: Thanks for your question. Those metabolites were quantified in different transgenic lines and the PII was native. We provided a detailed description in the revised manuscript.

- L437-444: the RNA-seq data was used for what? this is not clear through the manuscript.

Response: Thanks for your question. Here the RNA-seq data was used for gene annotation across the genome.

- L532-535: not clear if PII was fused to GFP which was used for the localization study and then used for metabolomics analysis or PII without GFP was used for the metabolomics analysis.

Response: Thanks for your question. Yes, the PII fused to GFP was used for the localization study and the native PII was used for the metabolomics analysis. We provided a detailed description in the revised manuscript and Supplementary Notes.

- L540: “relative PII expression was measured”, which Figure? not clear

Response: Many thanks for your question. In the revised manuscript, we provided the relative PII expression levels in the supplementary figures (Fig. S50).

Figure S50. Real-Time qRT-PCR of the *PIIs* in wide-type *PII*-expressing lettuce plants.

WT, wild-type lettuce plants; EV, empty vector control; Line 1~3, L1, L2, and L3 represent three independent transgenic lines;

- L556-565: if the authors used a PII-GFP fusion as I assume, how did they confirm that the PII is functional? – there is a lot of indications that the GFP could hinder the full function of PII, especially for the interaction with ACCase, due to the hindrance.

Response: Thanks for your question. As we described previously, the transgenic plants with At-PII are two types. One type is with GFP fusion since we would like to see whether the foreign PII gene functions in Chloroplast. Another type are not with GFP, which are used to do the physiological and metabolic tests. To demonstrate the function of these PIIs, we conducted a series of *in vitro* and *in vivo* tests (BiFC, Pull-down). Experiments have shown that PII of these three species can interact with NAGK in lettuce *in vitro* and *in vitro* (Figs. 3e, 3f, S51, S52). Therefore, we think the PII transformed into lettuce can play their biological function.

- L563: I am not familiar with this method to measure the ACCase activity, can you describe it briefly?

Response: Thank you very much for your question. We use Molybdenum blue method to analysis the ACCase activity. In brief, ACC can catalyze acetyl coenzyme A, NaHCO₃ and ATP to generate malonyl CoA, ADP and inorganic phosphorus. Molybdenum blue and phosphate generate a substance with a characteristic absorption peak at 660nm. ACC activity is determined by measuring the increase of inorganic phosphorus by ammonium molybdate phosphorus determination method. We measured the ACCase activity of WT and PII-Overexpressing lettuce using BioTek Synergy H1 Multimode Microplate Reader (Agilent, US) for three biological replicates and three technical replicates. We provided a detailed description in the revised manuscript.

Supplementary note:

- L1115-1116: I am not aware the PII influence on Glu and Gln biosynthesis, which ref. do you mean? and what do you mean by negative feedback of arginine? “Not clear”

Response: Thanks for your question. We are sorry for the misleading brief description of the PII function. We changed the sentences in the revised version and added the necessary references.

- L1111-1121: Can the authors confirm by any biophysical or biochemical methods that the lettuce NAGK is still interacting with AraPII? using pulldown or size exclusion chromatography or surface plasmon resonance or any relative method.

Response: Thank you very much for raising this suggestion. As we described previously, to demonstrate the function of these PIIs, we conducted a series of *in vitro* and *in vivo* tests (BiFC, Pull-down). Experiments have shown that PIIs of these three species can interact with NAGK in lettuce *in vitro* and *in vivo* (Figs. 3e, 3f, S51b,c, S52a,b). Therefore, we think the PIIs transformed into lettuce can play their biological function.

Summary

Taken together, we really appreciate all the insights on PII and PII signal machinery from the Reviewer and benefit a lot from the Reviewer's suggestions. Besides the transgenic study of At-PII in lettuce, following the Reviewer's suggestion, we added the studies of Dc-PII (carrot) and Sl-PII (tomato) in lettuce. Here, based on all experimental results, we would like to make a summary as follows.

- 1) It is for sure that all of At-PII, Dc-PII, and Sl-PII could interact with NAGK (from lettuce) based on *in vivo* and *in vitro* experiments.
- 2) Both the N (nitrate) uptake and the activity of ACCase are inhibited after overexpressing exogenous PIIs into lettuce, which are consistent in At-PII, Dc-PII, and Sl-PII transgenic plants. However, we don't know the reason why the nitrate uptake was repressed.
- 3) The contents of amino acids including glutamic acid, glutamine, and arginine are disturbed in transgenic plants. For Dc-PII and Sl-PII, all of them are increased. Instead, in At-PII transgenic plants, they are decreased. Our guess is that the truncated Q-loop in At-PII might be a reason, but not sure.
- 4) In terms of tens of millions years of evolution, the Asteraceae species should have evolved a distinct N-C balance system in the absence of PII. Currently, it is not clear how much/complicated change has occurred, even if we guess it should be much or complicated.

As the Reviewer mentioned that "this work may inspire future studies to investigate the metabolic adaptation machinery of Asteraceae on physiological and biochemical levels", we would like to emphasize more on our discoveries mainly based on omics studies considering the potential complexity of the unique N-C balance system in Asteraceae.

References

Baud S, Bourrellier ABF, Azzopardi M, Berger A, Dechorgnat J, Daniel-Vedele F, Lepiniec L, Miquel M, Rochat C, Hodges M, et al. 2010. PII is induced by WRINKLED1 and fine-tunes fatty acid composition in seeds of *Arabidopsis thaliana*. *Plant Journal* **64**: 291–303.

Bourrellier ABF, Valot B, Guillot A, Ambard-Bretteville F, Vidal J, Hodges M. 2010. Chloroplast acetyl-CoA carboxylase activity is 2-oxoglutarate-regulated by interaction of PII with the biotin carboxyl carrier subunit. *Proceedings of the National Academy of Sciences of the United States of America* **107**: 502–507.

Chellamuthu VR, Ermilova E, Lapina T, Lüddecke J, Minaeva E, Herrmann C, Hartmann MD, Forchhammer K. 2014. A widespread glutamine-sensing mechanism in the plant kingdom. *Cell* **159**: 1188–1199.

Ferrario-Méry S, Besin E, Pichon O, Meyer C, Hodges M. 2006. The regulatory PII protein controls arginine biosynthesis in *Arabidopsis*. *FEBS Letters* **580**: 2015–2020.

Ferrario-Méry S, Bouvet M, Leleu O, Savino G, Hodges M, Meyer C. 2005. Physiological characterisation of *Arabidopsis* mutants affected in the expression of the putative regulatory protein PII. *Planta* **223**: 28–39.

Hsieh MH, Lam HM, Van De Loo FJ, Coruzzi G. 1998. A PII-like protein in *Arabidopsis*: Putative role in nitrogen sensing. *Proceedings of the National Academy of Sciences of the United States of America* **95**: 13965–13970.

Mizuno Y, Berenger B, Moorhead GBG, Ng KKS. 2007a. Crystal structure of *Arabidopsis* PII reveals novel structural elements unique to plants. *Biochemistry* **46**: 1477–1483.

Mizuno Y, Moorhead GBG, Ng KKS. 2007b. Structural basis for the regulation of N-acetylglutamate kinase by PII in *Arabidopsis thaliana*. *Journal of Biological Chemistry* **282**: 35733–35740.

Smith CS, Weljie AM, Moorhead GBG. 2003. Molecular properties of the putative nitrogen sensor PII from *Arabidopsis thaliana*. *Plant Journal* **33**: 353–360.

Reviewer #2 (Remarks to the Author):

Review of Comparative Genomics Reveals a Unique Nitrogen-Carbon Balance System in Asteraceae

This manuscript presents the results of two newly sequenced genomes and provides a comparative genomics analysis with other Asteraceae genomes. They posit that a whole genome duplication event in early in the history of Asteraceae evolution set the stage for the evolution of a novel nitrogen-carbon balance system in the family. They demonstrate that PII, which has a significant role in sensing and regulating nitrogen-carbon signals was lost in the ancestor of Asteraceae and Goodeniaceae. They hypothesize that whole genome duplication

events in Asteraceae led to the expansion of high-affinity nitrate transporter genes and fatty acid biosynthesis genes thereby allowing the family to be more evolutionarily successful. The paper was thorough in general its methods and analyses with some points noted below. The work is very thorough, and the addition of functional analyses is noteworthy.

Response: We really appreciate all the comments from the Reviewer. Thank you for your recognition on the discovery and novelty of our study. Based on our understanding, we divided your major concerns into three related parts and replied/explained one by one. Hope our efforts resolve these concerns.

The major concern I had with the thesis is that the authors failed to discuss any other reasons in the main body of the manuscript (though it was touched on in the notes) why Asteraceae has been so successful and the current presentation of the work seems to hinge on this one aspect.

Response: This is a fantastic question and suggestion. Why we did not present or emphasize other critical reasons (also contributing to the Asteraceae success) is that we originally prepared our paper into the letter version of Nature Genetics, which is quite short (2000 words). Thus, many discoveries and insights were packed into Supplementary Notes as the Reviewer noticed. The short version was kept when the manuscript transferred from Nature Genetics to Nature Communications. In the revised version, we followed the form of Nature Communications and several parts are retrieved from Supplementary Notes.

We strongly agree with the Reviewer that the success of Asteraceae is the evolutionary result based on complicated and systematic changes. The nitrogen-carbon balance system in absence of PII plays an important role in this, but certainly not all. Constrained by complicated genomic features (e.g., large genome sizes and a high ratio of repeat sequences), limited numbers of genomic resources in the Asteraceae, there are few studies systematically analyzing the success of Asteraceae from a comparative genomic perspective (or a molecular perspective), and our study is just a beginning. One of the advantages of our study is the *Sc. taccada* genome, the representative plant of the Goodeniaceae family that is closely related to the Asteraceae, by which many side-by-side comparisons could be made. In addition to the loss of PII in both Asteraceae and Goodeniaceae, we also made the following discoveries and observations which potentially also contribute to the success of Asteraceae (more details in our revised manuscript and Supplementary Notes).

First of all, employed gene families from the *Sc. taccada* genome, we traced back the origin time of Asteraceae to ~80 MYA for the first time by molecular clock which is much earlier than the previous studies (Fig. 2a; Supplementary Note 4.1)(Barreda et al., 2015; Mandel et al., 2019; Zhang et al., 2021). Secondly, we proposed that genomic novelty based on genome duplications (WGT) could be a key to the success of Asteraceae given that many retained duplicate genes are in the critical stress-related pathways such as PMEs (Supplementary Note 4.2; Fig. S17). Thirdly, we observed that the genomes of Asteraceae species prefer to possess a high percentage of transposons (especially LTRs), potentially caused by the high copy number of (retro)transposon-associated genes. These regulatory DNA elements derived from (retro)transposable elements were involved in a wide range of biological functions such as BDR4 (Supplementary Note 5; Fig.

S22). Fourth, a large number of gene families are enlarged in the Asteraceae genomes including transcription factors (zinc finger, LIM-types, ect.) and fat acid biosynthesis (Supplementary Note 5; Fig. S22), and new regulatory pairs are formed like miRNAs (Asteraceae specific) and Feronia genes (Supplementary Note 6.3; Fig. S38). All aspects discussed above provide evidence for the potential drivers and impacts of genomic dynamics on species radiation and excellent environmental adaptation of Asteraceae.

Figure S17. Whole-genome triplication-related genomic features in Asteraceae. **a**, Whole-genome syntenology visualization of *Sc. taccada* versus coffee (*C. arabica*) and two Asteraceae species, lettuce (*La. sativa*) and artichoke (*C. cardunculus*), respectively. Red circles indicate a 1:1 relationship, while green and blue circles show a 1:3 relationship. **b**, Enriched gene ontology terms of the genes in the triplication retained regions (TRRs) of four representative Asteraceae species. **c**, Microsynteny visualization of the *ADS3* syntenic pairs in the TRRs of lettuce and *Sc. taccada*. **d**, Percentages of genetic/repeat regions in the TRRs and the whole genome in four selected Asteraceae species. A sliding window of 1 Mbp was used to calculate each data point. ****, $P < 0.00001$ as determined by two-tailed Student's *t*-test.

Figure S22. Diversification and dynamics of (retro)transposons from Asteraceae genomes.

a, Correlation between genome size and long terminal repeat (LTR) retrotransposons ($R^2 = 0.74$). Green and blue dots represent Asteraceae species and others, respectively, while green and blue lines are their corresponding best linear fits. **b**, Density distribution of solo/intact ratio of LTR retrotransposons in the representative Asteraceae and phylogenetic relatives. Solid lines represent Asteraceae genomes, while dashed lines represent others. **c**, Density distribution of insertion time of LTR retrotransposons in the Asteraceae species and *Scaevola taccada*. **d**, Sequence divergence distribution patterns of transposable element (TE) hits presented as a violin plot. The most recent LTR retrotransposon sequences of LTR retrotransposon families were selected as representative sequences to detect additional TE hits in the genomes. **e**, Phylogenetic analysis of the LTR retrotransposon sequences (*Ty1/Copia*) in the *Lactuca sativa* genome. The maximum-likelihood and unrooted phylogenetic trees were constructed based on 1,564 *Ty1/Copia* aligned sequences that corresponded to the reverse transcriptase domain without premature termination codon. **f**, Enriched (retro)transposon-related InterPro entries in the Asteraceae species. An enrichment analysis was performed based on the functional domains of all the genes across the 29 surveyed species. *P* values were derived from a hypergeometric test with Bonferroni

correction. **g**, A typical high-copy lineage-specific gene family with retrotransposon-related domains in Asteraceae. **h**, Micro-synteny visualization of *BDR4* syntelogs harboring transposon-related domains in Asteraceae species but not in *Sc. taccada*. **i**, Phylogenetic tree of the transposon-related *BDR4* syntelogs in the Asteraceae species, *Sc. taccada*, *Vitis vinifera*, *Coffea arabica*, and *Arabidopsis thaliana*. **j, k**, An example of transcription factor binding site (TFBS) gain by the insertion of LTR retrotransposons. The TFBS in *LsNRG2* of *La. sativa* was possibly introduced by a LTR retrotransposon after speciation from *Sc. taccada* (**j**) and caused significantly different expression profiles among species (**k**).

Figure S38. Evolution of representative gene families related to the characteristics of the Asteraceae. **a**, Divergence of the *R2R3-MYB* transcription factor gene family in Asteraceae. Asteraceae-specific clades are marked with coloured lines. **b**, The *FERONIA* family in Asteraceae. The bar chart illustrates the number of genes for each *FERONIA* type (with or without the typical malectin-like domain) in all 29 investigated species. The phylogenetic tree includes the typical

FERONIA genes and Asteraceae-specific clades with the phloem protein 2-like (PP2L) domain. The unique miRNA-target pair (Asteraceae-specific miRN518 and *LSA2272*, a representative *FERONIA* gene lacking the sequence encoding the malectin-like domain) are illustrated by sequence alignment. This regulatory pair was supported by a parallel analysis of RNA ends (PARE)-seq experiment. **c**, The simplified phylogenetic tree of Glycosyl hydrolase family 32 (GH32) protein in different species and schematic diagram of inulin biosynthesis. For the phylogenetic analysis, we used genes from the 29 investigated species in this study and genes encoding sucrose:sucrose 1-fructosyltransferase (1-SST), fructan:fructan 1-fructosyltransferase (1-FFT), and fructan 1-exohydrolases (FEHs) previously identified in chicory (*Cichorium intybus*) and barley (*Hordeum vulgare*). The unique 1-SST and 1-FFT clades in Asteraceae species are shown in red and blue, respectively. **d**, Schematic diagram of the role of strictosidine synthase (STR) in indole alkaloid biosynthesis and the phylogenetic tree of identified genes of the *Beta-glucosidase gene family 1* in different species. The Asteraceae-specific clade is illustrated by dotted lines. **e**, The simplified phylogenetic tree of typical genes encoding pectinesterases highlighting Asteraceae-specific clades (purple lines).

It is striking and impactful that there are no PII genes and the paper offers a very cool demonstration of how the gene could have been lost. And the expansion of specific classes of genes is also compelling, but still the argument correlative and overreaches in its current state. Is there evidence that such a nitrogen-carbon balance system would be a priori hypothesized to lead to ecological and evolutionary success. This was not addressed in the otherwise very thorough notes either. If so, the strength of this story would be greatly enhanced.

Response: It is really appreciated that the Reviewer's comments on PII discovery. Nitrogen (N) and carbon (C) are essential for life, and their availability is often a limiting factor for plant growth in natural ecosystems. Cellular N/C balance in plants is finely coordinated by regulatory genes to sustain optimal growth and development. As the vital gene in regulating the N/C balance, the PII gene acts as the reporter of the C metabolic state of the cell by interdependently binding ATP/ADP and 2-oxoglutarate (2-OG) (Chellamuthu *et al.*, 2014). The PII-mediated N/C balance is critical for sustaining optimal growth and development.

As the Reviewer #1 suggested, to further confirm the function of PII in lettuce, beside At-PII (Arabidopsis), we also transformed Dc-PII (carrot) and Sl-PII (tomato) into lettuce, respectively, and followed a series of physiological tests. Even not all of the results are consistent in different transgenic plants, the representatives of amino acids, nitrate content, etc., are strongly disturbed, suggesting the substantial function of external PII in lettuce. These results also suggest there should exist a new nitrogen-carbon balance system in lettuce (Asteraceae).

Efficient N assimilation systems is critical for plants to survive in severe habitats or compete for nourishment. There are several analogies in the previous studies. As we mentioned in our manuscript, orchids are one of the very few flowering plant lineages that have been able to successfully colonize epiphytic or lithophytic niches, clinging to trees or rocks and growing in dry conditions using crassulacean acid metabolism (Zhang *et al.*, 2017), potentially resulting in their rich species diversity. Another example is Leguminosae which can fix atmospheric N through a symbiotic association with soil bacteria and have become widespread through the most spectacular radiations (Azani *et al.*, 2017). In our model, in absence of PII, the ability to absorb nitrogen (especially in low N condition) and synthesize carbon through fatty acid synthesis has

been strengthened (Fig. 6), which is essential for Asteraceae plants to survive in nutrient-limited environments.

Again, we would like to emphasize that this unique N-C balance system might take tens of million years to be formed considering the history of Asteraceae. Our study is just a beginning, more systematic inferences and experimental results are expected to further elucidate the unique N/C system.

What about lineages within Asteraceae that are not successful, for example, the Barnadesioideae. Additionally, Barker et al. show a duplication even shared with Calyceraceae and Asteraceae, plus the one that is unique to Asteraceae. This lack of sampling here means there is a missing link with Calyceraceae. While it could still be the Asteraceae-specific WGT, there should be more discussion of the lack of sampling and what that means for the study conclusions.

Response: The Reviewer posted a critical and interesting question. It is for sure that the genomes of Calyceraceae and other subfamilies of Asteraceae (like Barnadesioideae) will provide more insights on the unique nitrogen-carbon balance system and its potential roles in species diversity. The Calyceraceae family and basal subfamilies of Asteraceae including Barnadesioideae, Famatnanthoideae, and Stifftioideae (all three have small numbers of species and only distributed in South America) are of great significance for the study of species trait formation and differentiation in Asteraceae, but unfortunately there is currently no genome available. Therefore, we can only conclude based on the current genome and related data, such as the discussion of polyploidy, we clearly know Asteraceae species did not share the WGT event with *Sc. taccada* (transcriptome data show the same WGT occurred in both Asteraceae and Calyceraceae as the Reviewer mentioned (Barker *et al.*, 2016).

In addition, the diversity distribution of species is also closely related to the environment. The species in several subfamilies of Asteraceae are found only in South America or in a quite limited region (some occurring in extreme environments). Like some Asteraceae plants that can invade other places, if these species were grown in a different environment, there is also the possibility that they can quickly occupy the ecological niche and become a new invasive species. In our revised version, we added more relevant discussions related to this topic.

Please also note that I am not an expert in the nitrogen-carbon balance system, so while I found their functional work compelling, I can't speak in depth to the details and the rationale behind their conclusions drawn.

Response: To test PII function in lettuce, as suggested by the Reviewer #1, we supplemented more transgenic studies and subsequent physiologic experiments. All of them indicate that PII's join can make a huge impact, and further suggest a unique N/C balance system in Asteraceae species.

Additional Comments:

Lines 41-44: This statement is not true as per their phylogeny in Figure 1. “Here, we generated high-quality genome assemblies for stem lettuce (*Lactuca sativa* var. *angustana*), a member of the Asteraceae, and beach cabbage (*Scaevola taccada*), a representative of the Goodeniaceae family, which is phylogenetically the closest outgroup to the Asteraceae (Fig. 1a,b).”

Response: Thanks for your comment. From the phylogenetic tree, the closest family to Asteraceae is Calyceraceae, not Goodeniaceae. However, several studies (Kim *et al.*, 2005; Pozner *et al.*, 2012; Katinas *et al.*, 2016) suggest that Asteraceae and Calyceraceae are sister lineages. Goodeniaceae are the closest outgroup of both Asteraceae and Calyceraceae. That is the reason why we state “the Goodeniaceae family, which is phylogenetically the closest outgroup to the Asteraceae”.

Lines 62-73: Are there some typos in this paragraph. WGT-2 is not cited but seems to be discussed? As written, the description of triplication events is unclear.

Response: Many Thanks for these good catches. We mentioned the three polyploidization events in this paragraph: 1) The ancestral whole-genome triplication of the Eudicots (WGT- γ). 2) Whole-genome triplication (WGT-1) (common with Asteraceae species). 3) a lineage-specific whole-genome duplication in the sunflower genome (WGD-2). We added the related reference in the revised manuscript.

Line 430, where is the customized repeat library available? GitHub?

Response: The customized repeat library has been uploaded to Github and the link is https://github.com/maypolefly/lettuce_data . We added this link in the revised manuscript.

Line 462, was model selection used to choose this model?

Response: The model selection process was conducted using the ModelFinder software. We added the description in the revised method part.

Line 467, where are the gene trees or matrices used to generate these phylogenies?

Response: Thank you very much for your question. The gene trees have been uploaded to Github and the link is https://github.com/maypolefly/lettuce_data. We added this link in the revised manuscript.

Line 476, where are the data on the fossils that were used?

Response: Three fossil calibrations corresponding to the crown groups of angiosperms (~126 Mya), eudicots (~120 Mya), and monocots (~113 Mya) were implemented as minimum age constraints in our penalized likelihood dating analysis.

Line 535, what are the tissue culture/regeneration methods/details?

Response: Thank you very much for your question. We added a more detailed description about the culture/regeneration in the method part in the revised manuscript.

Line 563, cite manufacturer details.

Response: Great catch. The manufacturer details have been added.

Line 569, the link does not work. Please ensure all these data that are stated are deposited.

Response: Thanks very much for your question. The sequencing data used in this study, assembled chromosomes, unplaced scaffolds, and annotations have been deposited into the Genome Sequence Archive (GSA) and Genome Warehouse (GWH) database in the BIG Data Center (<https://bigd.big.ac.cn/gsa/index.jsp>) under Accession Number PRJCA007442. Annotated information on stem lettuce in detail can also be found in LettuceGDB (<https://lettucegdb.com/>).

Figure 2a, readability would be improved by ordering the nodes as is more traditional in phylogenies.

Response: Great suggestion. We re-ordered the nodes and re-drawn the figure by following more traditional in phylogenies.

Figure 2b, consider reorganizing taxa list perhaps phylogenetically.

Response: We have revised Figure 2b accordingly.

Figure 4e, the colors corresponding to the taxa look different from the legend, e.g, legend H. annuus is darker than in the figure.

Response: The figure was revised accordingly.

There are 39 supplemental tables but they are not mentioned in the body of the manuscript.

Response: Thank you very much for your question. To be able to provide a more complete and sufficient description of the data, we also provided supplemental tables, figures, and notes. All the detailed descriptions of the data could be found in the supplementary notes. As you

suggested, we cited all of the supplementary tables and notes in the revised manuscript. We also respect the requirements of the journal and make format adjustments.

Data for the selection analyses on the genes is absent from the manuscript and the supplementals, please include.

Response: Many thanks for your question. The description for the selection analyses on the genes was added in the revised manuscript.

References

Azani N, Babineau M, Bailey CD, Banks H, Barbosa AR, Pinto RB, Boatwright JS, Borges LM, Brown GK, Bruneau A, et al. 2017. A new subfamily classification of the leguminosae based on a taxonomically comprehensive phylogeny. *Taxon* **66**: 44–77.

Barker MS, Li Z, Kidder TI, Reardon CR, Lai Z, Oliveira LO, Scascitelli M, Rieseberg LH. 2016. Most compositae (Asteraceae) are descendants of a paleohexaploid and all share a paleotetraploid ancestor with the calyceraceae. *American Journal of Botany* **103**: 1203–1211.

Barreda VD, Palazzesi L, Tellería MC, Olivero EB, Raine JI, Forest F. 2015. Early evolution of the angiosperm clade Asteraceae in the Cretaceous of Antarctica. *Proceedings of the National Academy of Sciences of the United States of America* **112**: 10989–10994.

Chellamuthu VR, Ermilova E, Lapina T, Lüddecke J, Minaeva E, Herrmann C, Hartmann MD, Forchhammer K. 2014. A widespread glutamine-sensing mechanism in the plant kingdom. *Cell* **159**: 1188–1199.

Katinas L, Hernández MP, Arambarri AM, Funk VA. 2016. The origin of the bifurcating style in Asteraceae (Compositae). *Annals of Botany* **117**: 1009–1021.

Kim KJ, Choi KS, Jansen RK. 2005. Two chloroplast DNA inversions originated simultaneously during the early evolution of the sunflower family (Asteraceae). *Molecular Biology and Evolution* **22**: 1783–1792.

Mandel JR, Dikow RB, Siniscalchi CM, Thapa R, Watson LE, Funk VA. 2019. A fully resolved backbone phylogeny reveals numerous dispersals and explosive diversifications throughout the history of Asteraceae. *Proceedings of the National Academy of Sciences of the United States of America* **116**: 14083–14088.

Pozner R, Zanotti C, Johnson LA. 2012. Evolutionary origin of the asteraceae capitulum: Insights from calyceraceae. *American Journal of Botany* **99**: 1–13.

Zhang C, Huang CH, Liu M, Hu Y, Panero JL, Luebert F, Gao T, Ma H. 2021. Phylotranscriptomic insights into Asteraceae diversity, polyploidy, and morphological innovation. *Journal of Integrative Plant Biology* **63**: 1273–1293.

Zhang GQ, Liu KW, Li Z, Lohaus R, Hsiao YY, Niu SC, Wang JY, Lin YC, Xu Q, Chen LJ, et al. 2017. The *Apostasia* genome and the evolution of orchids. *Nature* **549**: 379–383.

Reviewer #3 (Remarks to the Author):

In the study entitled “Comparative Genomics Reveals a Unique Nitrogen-Carbon Balance System in Asteraceae”, Shen and colleagues constructed two chromosome-scale genomes for lettuce (*Lactuca sativa* var. *angustana*, a member of the Asteraceae) and *Scaevola taccada*, (a member of the closest outgroup, the Goodeniaceae). They further performed comparative genomics analysis for 29 representative terrestrial plant species, and deduced that Asteraceae was originated from the paleopolyploidization event which occurred ~80 MYA. Notably, the detail comparative genomics analysis revealed that the Asteraceae genomes absence PII, the universal regulator of nitrogen-carbon (N-C) assimilation present in almost all domains of life. They thus proposed that the Asteraceae evolved a unique N-C balance system following the loss of PII, resulting in enhanced N uptake capacity and fatty acid biosynthesis. This study has groundbreaking for the evolution of Asteraceae. The manuscript was well organized. I only have few comments:

Response: Thanks very much for your positive comments. As you said, to decipher the genetic basis of Asteraceae with high biodiversity and excellent adaptability, we provided insights through genomic comparisons of 29 representative terrestrial plant species, including two new chromosome-scale genome assemblies of lettuce and *Scaevola taccada*. The origin of Asteraceae was traced back to ~80 million years ago and the accompanying paleopolyploidization was predicted to provide a foundation for adaptive evolution. The dynamic repetitive elements drove the diversification of Asteraceae genomes and impacted their regulatory or genetic regions. Systematic comparative genomics showed genes associated with adaptation in Asteraceae have been amplified and neo-functionalized. The core regulator of carbon-nitrogen assimilation, PII, was lost across Asteraceae, and a new carbon-nitrogen balance system provides a solid molecular basis for the adaptability of Asteraceae. According to your suggestions, we have adjusted the general structure of the article and enriched the content, and hope our modification can meet your satisfaction.

1. The basic information of assembled genome should be presented in the main text, for example: “Line 46-48: “The genome assembly of *Sc. taccada* (ST1.0) contained 8 pseudo-chromosomes and covered 1,159 Mb with detailed annotations (Extended Data Fig. 2; Supplementary Note 3).” The annotated gene number and the length of N50 should be presented.

Response: Many thanks for your great suggestion. We originally prepared our paper into the letter version of Nature Genetics, which is quite short (2000 words). Thus, many results, discoveries, and discussions were packed into Supplementary notes as the Reviewer noticed. The short version was kept when the manuscript was transferred from Nature Genetics to Nature Communications. In the revised version, we followed the form of Nature Communications, and several parts are retrieved from Supplementary Notes. In the revised manuscript, we gave a detailed description of both two high-quality genome assemblies including the gene number and the length of N50.

2. Fig. 3b, please label the centromere position on chromosomes.

Response: That is a great catch. We have labeled the candidate centromere position on chromosomes in the revised Fig. 3b.

3. Extended Data Fig. 2, No. of Sequences should be No. of contigs. HiC should be Hi-C. Effective number should be kept consistent, for example: Complete BUSCO (%) 95.42 (*La. sativa*) and 95.2 (*Sc. taccada*).

Response: Thanks for these catches. We adjusted the words and make them consistent throughout the article.

The author may need to reformat for Nature communications. I would suggest the editor to accept the manuscript after mirror revision.

Response: Thank the Reviewer for this suggestion. We have reformatted the manuscript based on the requirement of Nature communications.

Reviewers' Comments:

Reviewer #1:

Remarks to the Author:

As I wrote earlier, this is a fascinating study. Well designed and performed, providing us with an in depth understanding of the plant "Asteraceae" adaptation/evolution with potential role(s) of PII signaling proteins. In response to my comment/question raised on the earlier version, they extended the study further by adding a new aspect via expressing transgenically carrot/tomato PIIs, which contain intact Q-loop, into lettuce. They also extended/modified the text in several places as proposed. The authors have adequately responded to my concerns, and I am absolutely satisfied with the present version.

Minor points are as following:

- I am not familiar with fluorescence complementation (BiFC) analysis, so, may the authors explain it for better follow?
- In line 240: they mentioned "disturbed" without explaining the amino acids in case of carrot and tomato PIIs were significantly higher, while with At-PII were reduced. So, I would recommend to include part of section "7.2 Biological influence of the loss of PII in the Asteraceae family" into the main text. Can you also discuss briefly why At-PII could act differently from carrot and tomato PIIs.
- Please extend the discussion of PII section as in the rebuttal.

Reviewer #2:

Remarks to the Author:

I have reviewed the revised version of this manuscript. While I still agree that the interesting findings related to lineage specific lost genes in the Asteraceae PII genes are very cool, there are a number of areas in the manuscript where I feel the authors have not cited the literature properly or they have oversold their findings. I urge the authors to focus on the story about PII and the novel duplications and not dilute their work with stories and statements that have already been described and are well-known in the family's evolutionary history. In addition, the authors have some information incorrect in their manuscript regarding the evolutionary relationships and how to interpret outgroups, this along with overselling points that have been well-known leads me to ask for these items to be corrected so that the presentation of the data represents what is known from the field. A significant caveat here, is that Calyceraceae is an outgroup to Asteraceae and it is the closest outgroup family. The authors seemed to have misinterpreted what 'sister lineage' and 'outgroup' mean in citing some papers saying Goodeniaceae is the closest outgroup. Sister lineage is a relative term. The family Goodeniaceae is the sister lineage to Calyceraceae + Asteraceae. Therefore, this work would be stronger using the closest outgroup, which also does happen to share the genome duplication, but it is not widespread geographically and we don't have the status of PII lineage specific loss. I would ask that they state in their manuscript that the closest relative sharing the genome duplication is not available for study so their results should be interpreted with that caveat.

Also for example, statements such as these: "We traced the origin of Asteraceae to ~80 million years ago, and predicted the accompanying paleopolyploidization had a foundation role of adaptive evolution." This has long been presented in numerous papers some cited by the authors Barker et al. 2008; 2016, Badouin et al 2017, Zhang et al. 2021, etc.

In addition, key references are missing making the current study sounds like it was the first to present such results, e.g., repetitive data: Staton and Burke (2015) published a study on evolutionary trends in the family and this is not cited; and fatty acids evolution: Chapman and Burke 2012.

In the letter response to reviews, the authors said this: "First of all, employed gene families from the

Sc. taccada genome, we traced back the origin time of Asteraceae to ~80 MYA for first time by molecular clock which is much earlier than the previous studies (Fig. 2a; Supplementary Note 4.1)(Barreda et al., 2015; Mandel et al., 2019; Zhang et al., 2021).” This is incorrect and is somewhat confusing in the body of the manuscript. I urge the authors to not focus on this and even remove the reference to this, as it is not new information and has been accept for years that the family is old.

Minor point for main body:

Hairy pappus or pappi is not a term one usually uses, I suggest pappus of bristles or something similar

Minor points for supplementals:

This sounds like there were three genomes sequenced reword for clarity:

Page 3, line 93 “To solve these puzzles, we generated two high quality genome assemblies of stem lettuce (*Lactuca sativa* var. *angustana*), a representative economic crop of the Asteraceae family , and *Scaevola taccada* , a representative plant of the Goodeniaceae family that is phylogenetically the closest outgroup to Asteraceae (Fig s 1 a b).”

I tried to see the details of the organellar genomes sequenced, but I didn’t find them in the supplementals.

Page 22, line 553, how were orthologs assigned, Orthofinder as in section 7?

Reviewer #3:

Remarks to the Author:

The authors have addressed all my concerns and I would like to suggest the editor to accept the manuscript.

Reviewer #1 (Remarks to the Author):

As I wrote earlier, this is a fascinating study. Well designed and performed, providing us with an in depth understanding of the plant “Asteraceae” adaptation/evolution with potential role(s) of PII signaling proteins. In response to my comment/question raised on the earlier version, they extended the study further by adding a new aspect via expressing transgenically carrot/tomato PIIs, which contain intact Q-loop, into lettuce. They also extended/modified the text in several places as proposed. The authors have adequately responded to my concerns, and I am absolutely satisfied with the present version.

Response: Again, thanks a lot for your appreciation on our discovery. Your professionalism and in-depth knowledge of the field of PII research has greatly helped us improve our work.

Minor points are as following:

- I am not familiar with fluorescence complementation (BiFC) analysis, so, may the authors explain it for better follow?

Response: Bimolecular Fluorescence Complementation (BiFC) is a convenient method for verifying protein interactions in plants. We summarized more details as follows.

The BiFC assay is based on structural complementation between two non-fluorescent N-terminal and C-terminal fragments of a fluorescent protein, e.g., green fluorescent protein (GFP), yellow fluorescent protein (YFP). For BiFC analysis, earlier studies have showed that GFP can be split at a loop or within a β -stand. The split two fragments are non-fluorescent, but can then be fused to proteins of interest that may interact. If the proteins interact, the non-fluorescent fragments are brought into closely proximity and reconstitute a complete fluorescent protein that can be detected using fluorescence microscope or confocal microscope (as the following figure displayed).

Figure from Yutaka Kodama and Chang-Deng Hu, 2018 (<https://doi.org/10.2144/000113943>) (Kodama & Hu, 2012).

Scientists usually use transient expression methods to transfer the vectors encoded proteins of interest fused to the N-terminal and C-terminal fragments of GFP/YFP into the leaves of plants by *Agrobacterium tumefaciens*, then detect the interactions by confocal microscopy. This allows not only to observe whether the proteins interact in vivo, but also to determine the intracellular location of their interactions.

In line 240: they mentioned “disturbed” without explaining the amino acids in case of carrot and tomato PIIs were significantly higher, while with At-PII were reduced. So, I would recommend to include part of section “7.2 Biological influence of the loss of PII in the Asteraceae family” into the main text. Can you also discuss briefly why At-PII could act differently from carrot and tomato PIIs.

Response: Many thanks for your suggestions. As you suggested, we added the experimental results (different effects on the amino acid content) when expressing different exogenous PII in lettuce, and explained the potential reason in the PII part (highlighted in the revised manuscript).

- Please extend the discussion of PII section as in the rebuttal.

Response: We thank you very much for this great suggestion. The discussion of PII section was extended and many points as in the rebuttal were included (highlighted in the revised manuscript).

References

Kodama Y, Hu CD. 2012. Bimolecular fluorescence complementation (BiFC): A 5-year update and future perspectives. *BioTechniques* **53**: 285–298.

Reviewer #2 (Remarks to the Author):

I have reviewed the revised version of this manuscript. While I still agree that the interesting findings related to lineage specific lost genes in the Asteraceae PII genes are very cool, there are a number of areas in the manuscript where I feel the authors have not cited the literature properly or they have oversold their findings. I urge the authors to focus on the story about PII and the novel duplications and not dilute their work with stories and statements that have already been described and are well-known in the family’s evolutionary history.

Response: Again, many thanks for your comments, and we really appreciate your recognition on the novelty of our work on PII. The manuscript has been improved a lot with your professional insights in the last revision. We strongly agree with you to focus on the story about PII. In fact, we organized most of results on genome assembly, WGT, comparative genomics, gene family and LTR family analysis into the supplementary notes. In the new revised version of the manuscript, we added more experimental details and discussion on the PII story (also suggested by another Reviewer). With regard to the evolutionary history of Asteraceae as you mentioned, we tried our best to use the accurate terms, cite the proper literatures and correct the incorrect ones in both manuscript and supplementary notes. Again, thanks a lot for your professional suggestions, and we

hope you find these revisions satisfactory.

In addition, the authors have some information incorrect in their manuscript regarding the evolutionary relationships and how to interpret outgroups, this along with overselling points that have been well-known leads me to ask for these items to be corrected so that the presentation of the data represents what is known from the field. A significant caveat here, is that Calyceraceae is an outgroup to Asteraceae and it is the closest outgroup family. The authors seemed to have misinterpreted what ‘sister lineage’ and ‘outgroup’ mean in citing some papers saying Goodeniaceae is the closest outgroup. Sister lineage is a relative term. The family Goodeniaceae is the sister lineage to Calyceraceae + Asteraceae. Therefore, this work would be stronger using the closest outgroup, which also does happen to share the genome duplication, but it is not widespread geographically and we don’t have the status of PII lineage specific loss. I would ask that they state in their manuscript that the closest relative sharing the genome duplication is not available for study so their results should be interpreted with that caveat.

Response: Many thanks for the Reviewer’s clarification on “sister lineage” and “outgroup”. With your suggestions, we have made corresponding modifications. For instance, we described the evolutionary relationship of Goodeniaceae, Calyceraceae and Asteraceae more accurately: "the Goodeniaceae family that is the sister lineage to Calyceraceae and Asteraceae”.

Meanwhile, we strongly agree with the Reviewer. Because currently the study on Calyceraceae and basal subfamilies of Asteraceae including Barnadesioideae, Famatnanthoideae, and Stiffioideae, is quite limited. Our conclusions are only based on the current genome and related data. In the revised version, we extended the discussion on this topic (Line 373-377, highlighted in the revised manuscript).

Also for example, statements such as these: “We traced the origin of Asteraceae to ~80 million years ago, and predicted the accompanying paleopolyploidization had a foundation role of adaptive evolution.” This has long been presented in numerous papers some cited by the authors Barker et al. 2008; 2016, Badouin et al 2017, Zhang et al. 2021, etc.

Response: Thank you for raising this issue. As you mentioned, similar results have been reported by others with different data/methods, including transcriptomes (Barker *et al.*, 2016; Mandel *et al.*, 2019; Zhang *et al.*, 2021) and fossil evidences (Mandel *et al.*, 2019). The difference in our paper is we employed the first available genome from Goodeniaceae, *Sc. taccada* genome, and from a whole genome comparison, we made the same conclusions. Thus, we adjusted the sentence into “We estimated the origin of Asteraceae to ~80 million years ago, and predicted the accompanying paleopolyploidization had a foundation role of adaptive evolution”. Besides, we added the necessary references in the main txt (e.g., Line 162, Line 299 in the revised manuscript).

In addition, key references are missing making the current study sounds like it was the first to present such results, e.g., repetitive data: Staton and Burke (2015) published a study on evolutionary trends in the family and this is not cited; and fatty acids evolution: Chapman and Burke 2012.

Response: We have added these key references in discussion part of both revised manuscript (Line 333) and supplementary notes (Line 684).

In the letter response to reviews, the authors said this: “First of all, employed gene families from the *Sc. taccada* genome, we traced back the origin time of Asteraceae to ~80 MYA for first time by molecular clock which is much earlier than the previous studies (Fig. 2a; Supplementary Note 4.1) (Barreda et al., 2015; Mandel et al., 2019; Zhang et al., 2021).” This is incorrect and is somewhat confusing in the body of the manuscript. I urge the authors to not focus on this and even remove the reference to this, as it is not new information and has been accepted for years that the family is old.

Response: Thanks a lot for your professional advice. We have followed your suggestions and made corresponding changes. As we mentioned above, we would like to emphasize that with the first available genome from Goodeniaceae (*Sc. taccada* genome) and from a whole genome comparison, we confirmed the previous discovery.

Minor point for main body:

Hairy pappus or pappi is not a term one usually uses, I suggest pappus of bristles or something similar

Response: Thank you for your professional suggestion, we have made the change as you suggested (highlighted in the revised manuscript).

Minor points for supplementals:

This sounds like there were three genomes sequenced reword for clarity:

Page 3, line 93 “To solve these puzzles, we generated two high quality genome assemblies of stem lettuce (*Lactuca sativa* var. *angustana*), a representative economic crop of the Asteraceae family, and *Scaevola taccada*, a representative plant of the Goodeniaceae family that is phylogenetically the closest outgroup to Asteraceae (Fig s 1 a b).”

Response : Thanks for your suggestion. We have reworded this sentence (highlighted in the revised manuscript and supplementary file).

I tried to see the details of the organellar genomes sequenced, but I didn't find them in the supplementals.

Response: Many thanks for raising this issue. We have supplied figures in the supplementary notes, and added the relevant text description (Line 230-233, 443-448). Besides, all the sequences could be found in the Github (https://github.com/maypoleflyn/lettuce_data).

Page 22, line 553, how were orthologs assigned, Orthofinder as in section 7?

Response: Synteny comparisons were identified by MCscan with default parameters to predict paralogs and orthologs. More details are in the methods of manuscript. In the revised

supplementary notes, we also added more details on methods (Line 558).

References

Barker MS, Li Z, Kidder TI, Reardon CR, Lai Z, Oliveira LO, Scascitelli M, Rieseberg LH. 2016. Most compositae (Asteraceae) are descendants of a paleohexaploid and all share a paleotetraploid ancestor with the calyceraceae. *American Journal of Botany* **103**: 1203–1211.

Mandel JR, Dikow RB, Siniscalchi CM, Thapa R, Watson LE, Funk VA. 2019. A fully resolved backbone phylogeny reveals numerous dispersals and explosive diversifications throughout the history of Asteraceae. *Proceedings of the National Academy of Sciences of the United States of America* **116**: 14083–14088.

Zhang C, Huang CH, Liu M, Hu Y, Panero JL, Luebert F, Gao T, Ma H. 2021. Phylotranscriptomic insights into Asteraceae diversity, polyploidy, and morphological innovation. *Journal of Integrative Plant Biology* **63**: 1273–1293.